# The Seasonal-to-Multiyear Large Ensemble (SMYLE) Prediction System using the Community Earth System Model Version 2

Stephen G. Yeager[1], Nan Rosenbloom[1], Anne A. Glanville[1], Xian Wu[1], Isla Simpson[1], Hui Li[1], Maria J. Molina[1], Kristen Krumhardt[1], Samuel Mogen[2], Keith Lindsay[1], Danica Lombardozzi[1], Will Wieder[1], Who M. Kim[1], Jadwiga H. Richter[1], Matthew Long[1], Gokhan Danabasoglu[1], David Bailey[1], Marika Holland[1], Nicole Lovenduski[2], Warren G. Strand[1], Teagan King[1]

[1]National Center for Atmospheric Research, Boulder, Colorado, USA
[2]Department of Atmospheric and Oceanic Sciences, University of Colorado, Boulder, Colorado, USA

*Correspondence to*: Stephen G. Yeager (yeager@ucar.edu)

**Abstract.** The potential for multiyear prediction of impactful Earth system change remains relatively underexplored compared to shorter (subseasonal to seasonal) and longer (decadal) timescales. In this study, we introduce a new initialized prediction system using the Community Earth System Model Version 2 (CESM2) that is specifically designed to probe potential and actual prediction skill at lead times ranging from 1 month out to 2 years. The Seasonal-to-Multiyear Large Ensemble (SMYLE) consists of a collection of 2-year long hindcast simulations, with 4 initializations per year from 1970 to 2019 and an ensemble size of 20. A full suite of output is available for exploring near-term predictability of all Earth system components represented in CESM2. We show that SMYLE skill for El Niño-Southern Oscillation is competitive with other prominent seasonal prediction systems, with correlations exceeding 0.5 beyond a lead time of 12 months. A broad overview of prediction skill reveals varying degrees of potential for useful multiyear predictions of seasonal anomalies in the atmosphere, ocean, land, and sea ice. The SMYLE dataset, experimental design, model, initial conditions, and associated analysis tools are all publicly available, providing a foundation for research on multiyear prediction of environmental change by the wider community.

## 1 Introduction

The desire for accurate advanced warning of high-impact, near-term, and regional environmental change has inspired rapid growth in the field of Earth system prediction using initialized coupled climate models. Recent developments in computing, modeling, and observing have made it possible to explore the potential for predictions that extend well beyond the two-week timescale of traditional weather forecasting. Multinational coordinated research efforts exist to advance the science of subseasonal (~3-4 weeks), seasonal (~12 months), and even decadal (~10 years) prediction of the physical, chemical, and biological components of the Earth system (Merryfield et al. 2020). Well-defined protocols for prediction system design have facilitated the use of large, multi-model ensembles that have, in some instances, revealed unexpectedly high potential

for skillful initialized prediction on climate timescales (e.g., Smith et al. 2020). The prevalence of unrealistically low signal-to-noise ratios in models routinely used in Coupled Model Intercomparison Project (CMIP) simulations means that low prediction skill does not necessarily imply a lack of potential for skillful prediction (Scaife and Smith 2018; Zhang et al. 2021). Without *a priori* knowledge of the inherent predictability limits of the multitude of Earth system processes at work on subseasonal to decadal timescales, continuous experimentation with ever more sophisticated models and methodologies is the sole path forward to advance our understanding of the scope for practicable Earth system prediction.

Prediction systems are generally designed to probe a limited range of timescales that are dominated by climate phenomena considered to be the key sources of predictability (Merryfield et al. 2020). While seamless prediction from weather to decadal timescales is an aspiration, practical considerations demand a judicious choice of hindcast simulation length, start frequency, and temporal coverage to provide robust statistics given limited resources. Thus, subseasonal systems designed to explore the predictability associated with the Madden-Julian Oscillation (MJO), sudden stratospheric warmings (SSWs), and the North Atlantic Oscillation (NAO) are comprised of ensemble simulations of order 1-month duration, initialized weekly over a span of roughly 17 years (Pegion et al. 2018; Richter et al. 2022). Seasonal prediction systems primarily focus on the climate impacts associated with the El Niño-Southern Oscillation (ENSO), and thus seasonal protocols call for ensemble simulations lasting up to 12 months, initialized monthly over the past 30 years (Becker et al. 2020). At the far end of the initialized climate prediction spectrum, decadal forecast systems require ensemble simulations of up to 10-year duration, initialized annually over a historical time window of 50 years or more (Boer et al. 2016) to sample different phases of low-frequency modes such as Atlantic Multidecadal Variability (AMV). The development of clear experimental protocols for exploring prediction on these different timescales has facilitated useful multi-model analyses and intercomparisons while also giving rise to distinct sub-groups within the Earth system prediction research community.

The potential for skillful prediction on interannual timescales remains much less well examined compared to other timescales, in part because the protocols for seasonal and decadal systems are not well-suited for assessing multiyear lead times, and because there is not a well-organized protocol focused on this timescale. Seasonal hindcasts are usually integrated for only 1 year, and while decadal hindcasts do encompass the multiyear timescale, they are only initialized once per year (usually on November 1) and so potentially important seasonality effects are missed. Previous work in this area has focused on exploring the potential for extended ENSO forecasts. Several studies have reported high skill at up to 2-year lead times for predictions of select multiyear La Niña events when hindcasts are initialized close to the preceding strong El Niño events (Luo et al. 2008; DiNezio et al. 2017; Wu et al. 2021). A large ensemble analysis of two decadal prediction systems (initialized in November) revealed modest potential for skillful climate prediction in some regions/seasons at lead times greater than 12 months, with notably enhanced skill during active ENSO periods (Dunstone et al. 2020). Such ENSO-related forecasts-of-opportunity could have significant practical utility if highly predictable initial states could be identified in advance with confidence.

In addition to ENSO, potential sources of predictability on seasonal to multiyear timescales include upper ocean heat content in the extratropics (Yeager et al. 2018), soil moisture (Esit et al. 2021), sea ice thickness (Koenigk and Mikolajewicz 2009), snow cover (Orsolini et al. 2013; Ruggieri et al. 2022), the quasi-biennial oscillation (QBO) of stratospheric winds (Butler et al. 2016), volcanic activity (Hermanson et al. 2020), greenhouse gas forcing (Doblas-Reyes et al. 2006; Boer et al. 2013), or

some combination thereof (Chikamoto et al. 2017). The sources of predictability for the biogeochemical (BGC) components of the Earth system remain unclear, but decadal prediction systems that include prognostic carbon cycle components have revealed promising potential to expand the scope of initialized prediction beyond the physical climate. Recent work has highlighted multiyear prediction skill for quantities such as air-sea $CO_2$ and terrestrial carbon fluxes (Lovenduski et al. 2019a; Lovenduski et al. 2019b; Ilyina et al. 2020), ocean acidification (Brady et al. 2020), ocean net primary productivity

(Krumhardt et al. 2020), and marine ecosystems (Park et al. 2019). These pioneering results merit closer examination in experiments that explicitly target the multiyear timescale. Recent work has identified robust multidecadal modulations of seasonal prediction skill (Weisheimer et al. 2017; O'Reilly et al. 2020), which suggests that a focused multiyear prediction framework could shed further light on important interactions between seasonal, interannual, and decadal processes (e.g., the state-dependence of multiyear predictability of seasonal climate) that would otherwise remain obscure. As a result of the

signal-to-noise paradox (Scaife and Smith 2018), a large ensemble size appears to be a prerequisite for skillful interannual predictions of some impactful atmospheric variations such as the NAO (Dunstone et al. 2016; Dunstone et al. 2020). A multiyear prediction protocol would permit the use of multi-model large ensembles that have been found to consistently outperform individual models in subseasonal (Richter et al. 2020) and seasonal (Becker et al. 2020) applications.

In this study, we introduce a new initialized prediction system--the Seasonal-to-Multiyear Large Ensemble (SMYLE)--that is specifically designed for exploring Earth system predictability out to 24-month lead times. The SMYLE collection of hindcasts is unprecedented in size (20-member ensembles initialized quarterly between 1970-2019) and scope (e.g., it includes a full suite of prognostic ocean and land biogeochemistry variables). SMYLE uses the latest version of the Community Earth System Model (CESM2; Danabasoglu et al. 2020) and is intended to be a foundational resource for future

prediction research by the CESM community. In addition to an extensive catalog of hindcast output from multiple components of the Earth system, the SMYLE data release includes historical state reconstructions for the ocean, sea ice, and land that were used for initializing component models. SMYLE is more than just a dataset, however, as it establishes an extensible experimental framework that will facilitate future CESM model and prediction system development activities. It is likely that the broader community impacts of SMYLE will be as significant as (or greater than) the results themselves as

community members build on this dataset with their own targeted experiments.

    The goals of this paper are to: 1) motivate the need for focused exploration of multiyear Earth system prediction; 2) document SMYLE experimental design; 3) provide a broad overview of SMYLE performance for various Earth system

quantities of interest; and 4) lay the foundations for and inspire broad community involvement in initialized prediction

research using CESM. A detailed description of the SMYLE experimental design follows in Section 2. Section 3 presents a brief survey of SMYLE skill for a variety of global and regional fields from each of the CESM2 component models. Section 4 offers some conclusions from this preliminary analysis and includes pointers to SMYLE data and code resources.

## 2 Experiment Description and Methods

SMYLE consists of a large collection of 24-month-long initialized hindcast simulations using the CESM2 model configured

at nominal 1° horizontal resolution in each of the component models. The atmosphere model is the Community Atmosphere Model version 6 (CAM6) with 32 vertical levels; the ocean model is the Parallel Ocean Program version 2 (POP2) with 60 vertical levels and prognostic ocean BGC using the Marine Biogeochemistry Library (MARBL; Long et al. 2021); the sea ice model is CICE version 5.1.2 (CICE5) with 8 vertical layers; and the land model is the Community Land Model version 5 (CLM5; Lawrence et al. 2019) with interactive biogeochemistry and agricultural management. The CESM2 overview paper

(Danabasoglu et al. 2020) provides additional details and references for readers interested in learning more about the component models. Hindcasts are initialized quarterly (1st of the month of November, February, May, and August) for each year between 1970 and 2019. Each hindcast includes 20 ensemble members, with ensemble spread introduced using a random field perturbation method at initialization, as is done in CESM subseasonal predictions (Richter et al. 2020, 2022). The complete set of SMYLE hindcasts therefore comprises 8,000 model simulation years corresponding to roughly 400 TB

of model output.

The historical initial conditions for the models used in SMYLE are derived from the Japanese 55-year Reanalysis (JRA-55; Kobayashi et al. 2015)—an ongoing (1958 to present) atmospheric data assimilation product that uses a relatively high-resolution atmospheric model (~55 km). The atmosphere model is initialized by directly interpolating the JRA-55 analysis

state onto the CAM6 model grid. The ocean and sea ice initial conditions are obtained from a forced ocean--sea-ice (FOSI) configuration of CESM2 that uses JRA55-do (Tsujino et al. 2018) atmospheric fields as surface boundary conditions, consistent with the protocol for version 2 of the Ocean Model Intercomparison Project (OMIP2; Griffies et al. 2016) of CMIP6. The SMYLE FOSI simulation consists of 6 consecutive cycles of 1958-2018 forcing, with the 6th cycle (used for SMYLE) extended through 2019 using extended JRA55-do forcing (Tsujino, pers. comm.). In addition to providing

historical physical states for the ocean and sea-ice that are compatible with each other as well as with the CESM2 model, the FOSI simulation yields a reconstruction of ocean BGC fields extending back to 1958. The land model is initialized from a forced (land-only) simulation of CLM5 using the CRU-JRAv2 forcing that was also used for Trends in land carbon cycle (TRENDY) simulations with CLM5 that were contributed to the Global Carbon Project (Friedlingstein et al. 2020). The CRU-JRAv2 data are applied cyclically from 1901-1920 to equilibrate the land state, with land carbon pools deemed close

enough to equilibrium after 4,000 years of spin-up (Fig. A1). A fully transient simulation following the TRENDY S3

protocol (Friedlingstein et al. 2020) was initialized from the spin-up run and integrated from 1901-2019 with forcings that include changes in climate, $CO_2$, land use and land cover, nitrogen deposition, and agricultural fertilization. Such land-only CLM5 simulations following the TRENDY protocol have been shown to be quite realistic for many but not all land variables, with CLM5 comparing well against other models for most fields (Friedlingstein et al. 2020, 2022). Furthermore, land-only CLM5 simulations show greatly improved realism compared to earlier CLM versions (Lawrence et al. 2019).

SMYLE FOSI is very similar, but not identical to, the CESM2 contribution to OMIP2 (Tsujino et al. 2020). First, to reduce model bias in sea-ice thickness and extent (particularly, in summertime) that were present in the CESM2 OMIP2 submission, larger values for sea-ice albedo were employed in SMYLE FOSI than in the default CESM2. Second, SMYLE FOSI used strong sea surface temperature (SST) restoring under sea-ice to maintain more realistic sea-ice thickness. Third, to improve the realism of BGC macronutrient profiles in the deep ocean, the non-dimensional ocean isopycnal diffusion lower bound parameter in SMYLE FOSI was reduced from 0.2 to 0.1. The first two modifications yielded significantly improved climatological Arctic sea-ice concentration in summer (Fig. A2) and year-round sea-ice thickness (Fig. A3) in SMYLE FOSI compared to OMIP2. Surface ocean fields remained largely unchanged by these sea-ice fixes, as shown for example by minor differences in upper ocean heat content bias (Fig. A4). In addition to improving deep BGC fields, the modification of the diffusion parameter greatly reduced the cooling drift in the deep ocean (below 2000m) that was present in the OMIP2 simulation (see Fig. 2 of Tsujino et al. 2020). This changed the global average temperature decrease over 6 forcing cycles from ~0.4°C to ~0.1°C (not shown). The modified sea-ice albedo and deep ocean diffusion settings were carried over to the fully-coupled SMYLE hindcast simulations. Other differences from the default CESM2 fully-coupled model configuration documented in Danabasoglu et al. (2020) include several modified parameter settings in MARBL and a crop-harvest bug fix in CLM5.

The historical radiative forcings used in SMYLE exactly match those used in members 51-100 of the CESM2 Large Ensemble (CESM2-LE; Rodgers et al. 2021). This 50-member ensemble largely follows the forcing protocols of CMIP6 (Eyring et al. 2016) for historical (1850-2014) and SSP3-7.0 (2015-2100) time periods. However, a significant deviation from the CMIP6 protocol is the use of smoothed biomass burning forcing during the 1990-2020 period to avoid spurious late 20th century warming associated with the introduction of satellite-based emissions forcings (Fasullo et al. 2021). While external forcings are identical, SMYLE differs from CESM2-LE members 51-100 in terms of: ocean deep diffusion parameter; sea-ice albedo settings; and MARBL tuning parameters, as listed above. Despite these differences, these CESM2-LE members should serve as a useful uninitialized benchmark for quantifying the benefits of initialization in SMYLE.

Results from the SMYLE hindcasts initialized in November (SMYLE-NOV) are compared to the 40-member CESM1 Decadal Prediction Large Ensemble (Yeager et al. 2018) that also used November 1st initialization (DPLE-NOV) but with starts only through 2017. Skill differences between SMYLE-NOV and DPLE-NOV are likely related to differences in

prediction system design that include: 1) ensemble size, 2) model physics (CESM2 vs. CESM1.1LENS), and 3) initialization methodology. Larger ensemble size tends to increase skill, with the magnitude of skill increase varying considerably from field to field (Scaife and Smith 2018; Yeager et al. 2018; Dunstone et al. 2020; Athanasiadis et al. 2020). We control for ensemble size effects in the SMYLE/DPLE comparison by randomly subsampling 20-member DPLE ensembles from the 40-member pool (repeated 100 times) and then taking the average of 20-member DPLE skill scores. SMYLE incorporates multiple developments in individual component models (Danabasoglu et al. 2020) as well as a more comprehensive initialization strategy (observation-based initialization of the atmosphere and land in addition to ocean and sea ice). It is not possible to attribute differences in SMYLE/DPLE skill to particular design choices, but it is nevertheless of interest to document similarities and differences that might be worth exploring in more depth with dedicated sensitivity experiments.

The primary focus of this general assessment of SMYLE performance is on hindcast skill for seasonally averaged fields (DJF, MAM, JJA, SON). Forecast lead time is defined here following common usage in seasonal prediction (e.g., Becker et al. 2020) with lead time (in months) denoting the interval between initialization and the start of a target season. For example, SMYLE hindcasts initialized in November (SMYLE-NOV) yield seasonal predictions at 7 lead times as follows: 1:DJF, 4:MAM, 7:JJA, 10:SON, 13:DJF, 16:MAM, 19:JJA. The terms "forecast season" or "forecast month" refer to the integer sequence of temporally averaged forecasts. Thus, the SMYLE-NOV hindcasts yield 7 forecast seasons as follows: 1:DJF, 2:MAM, 3:JJA, 4:SON, 5:DJF, 6:MAM, 7:JJA. Unless otherwise noted, drift is removed from forecasts by converting all fields to anomalies from a model climatology that varies with lead time. Observed anomalies are obtained by subtracting the equivalent climatology from the observational record. It is well known that long-term trends associated with external forcing contribute significantly to hindcast skill assessed over multidecadal time spans (e.g., Smith et al. 2019; Yeager et al. 2018). For simplicity and economy, this SMYLE skill assessment focuses primarily on detrended, interannual data to highlight the potential to predict multiyear departures from a linear trend (but skill for non-detrended data is also shown in Appendix B for select fields). Global hindcast fields (and corresponding observations) are mapped to a common regular grid (either 5°x5° for global fields or 3°x3° for land fields) using conservative mapping weights prior to skill score computation. This remapping is done to highlight aggregate regional skill, increase the efficiency of skill analysis, and improve the quality of global map plots that include significance markers (see Appendix B).

The primary skill metrics examined are the anomaly correlation coefficient (ACC) and normalized root mean square error defined (nRMSE) as follows:

$$ACC = \frac{\sum_{i=1}^{N} \acute{f}_i \, \acute{o}_i}{\sqrt{\sum_{i=1}^{N} \acute{f}_i^2 \, \sum_{i=1}^{N} \acute{o}_i^2}}$$

**( 1 )**

$$nRMSE = \frac{\sqrt{\frac{1}{N}\sum_{i=1}^{N}(\acute{f}_i - \acute{o}_i)^2}}{\sqrt{\frac{1}{N}\sum_{i=1}^{N}\acute{o}_i{}^2}} = \frac{RMSE}{\sigma_o} \qquad\qquad (2)$$

where $\acute{f}_i$ is the (dedrifted) ensemble mean forecast anomaly at verification time $i$, $\acute{o}_i$ is the corresponding anomaly from observations (or more generally, from the chosen verification dataset), $\sigma_o$ is the observed standard deviation, and $N$ is the temporal sample size. Unless otherwise indicated, $N$ is always maximized for a given pairing of forecast and verification time series. For example, $N = 50$ for SMYLE lead month 1 forecasts that verify in the interval 1970-2019 if observational data are available for that full period, but $N = 41$ if observations are only available from 1979-2019. The significance of ACC scores is determined from the p-value of a t-test that uses the effective sample size based on the autocorrelation of the time series (Bretherton et al. 1999). The null hypothesis of zero correlation is rejected if the two-tailed p-value is less than $\alpha = 0.10$, corresponding to a 90% confidence level. Significant differences between SMYLE-NOV and DPLE-NOV ACC scores for select fields are assessed based on p-values computed by comparing the single 20-member SMYLE-NOV score to the distribution of 20-member scores obtained from DPLE-NOV. The nRMSE metric reflects only the error variance (not the mean bias) and has the value of 1 for a zero anomaly (climatology) forecast.

The observational and/or reanalysis datasets used for skill verification are as follows: CRU-TS4.05 (Harris et al. 2020) for surface temperature over land; HadISST1 (Rayner et al. 2003) for surface temperature over ocean; GPCP version 2.3 for precipitation (Adler et al. 2018); ERA5 reanalysis for sea level pressure (Hersbach et al. 2020); OceanSODA-ETHZ (Gregor and Gruber 2021) for aragonite saturation state; National Snow & Ice Data Center (NSIDC) Sea Ice Index (Fetterer et al. 2017) for Arctic sea ice extent; Pan-Arctic Ice Ocean Modelling and Assimilation System (PIOMAS) for Arctic sea ice volume (Schweiger et al. 2011); and Best Track data for tropical cyclone activity (Knapp et al. 2010). We compare SMYLE ENSO skill to that of the North America Multimodel Ensemble system (NMME; Barnston et al. 2019).

## 3 Results

### 3.1 Global surface temperature and precipitation

The overall performance of SMYLE is summarized in global maps of ACC for surface temperature (Fig. 1) and precipitation (Fig. 2) as a function of start month (columns) and lead time (rows). As expected, skill is high in the first season following initialization and degrades with lead time. Surface temperature skill is generally higher over ocean than over land, but ACC scores exceeding 0.5 are evident for some land regions for lead times up to 13 months (e.g., northern South America). The following ocean regions stand out as hotspots of particularly high and long-lasting SST skill: the tropical Pacific and Atlantic, the subpolar North Atlantic, the western Indian, and the Pacific sector of the Southern Ocean. In each of these regions, significantly positive ACC scores (in places, exceeding 0.5) are found even at 19-month lead time (Fig. 1, bottom row). At long lead times, significantly positive land surface temperature skill is found primarily in low latitude regions

adjacent to the zones of high tropical ocean skill, while midlatitude land regions show low skill, in general. Possible exceptions are the west coast of North America (extending to Alaska) and the British Isles, which show significant ACC values (albeit less than 0.5) even in the second year of the predictions. For boreal summer (JJA), the most promising regions for multiyear prediction are the Caribbean and Central America, southeast Asia, western North America, Greenland, and parts of China and the Middle East. For boreal winter (DJF), statistically significant 2-year skill is found over Central America, Africa, and Australia. As expected, ACC scores are greatly enhanced almost everywhere when considering raw (non-detrended) data because of large amplitude, externally forced secular temperature trends (cf. Figs. 1, B1).

There is considerably less skill for seasonal precipitation than for surface temperature (cf. Figs. 1,2). Skill for precipitation is high over tropical oceans, particularly the tropical Pacific, for lead months 1-7, but it degrades faster with lead time than skill for surface temperature. Like for surface temperature, the high precipitation skill over tropical oceans generally does not extend to land regions except in select areas at short lead times. Potentially useful prediction skill (ACC>0.5) is seen for land precipitation over southwestern North America in DJF (lead month 1), Central America in JJA (lead months 1-4), Florida and adjacent island regions in DJF (lead months 1-7), Australia in JJA and SON (lead months 1-4), east Africa in DJF (lead months 1-4), and northern South America in all seasons (lead month 1) but particularly in DJF (lead months 1-10). The longest lasting precipitation skill is found over the Maritime Continent in the western Pacific where ACC in boreal spring (MAM) remains above 0.5 out to lead month 13 (and perhaps, lead month 16). Apart from that, there is scant evidence of useful skill for seasonal precipitation over either land or ocean in year 2 of SMYLE hindcasts, although many regions have ACC scores that are significantly positive at the 90% confidence level (Fig. 2, bottom 3 rows). Unlike surface temperature, precipitation ACC skill is largely insensitive to removal of a linear trend (cf. Figs. 2, B2).

The temperature and precipitation results from SMYLE are broadly in line with previous assessments of multiyear prediction skill (Dunstone et al. 2020). A key outstanding question is whether the low skill over land (particularly for precipitation) reflects fundamental predictability limits or, instead, potentially correctable flaws in model realism and/or prediction system design. We cannot answer that question, but we can at least explore the combined sensitivity to CESM model version and initialization methodology by comparing the skill of SMYLE-NOV to that of DPLE-NOV (rightmost two columns in Figs. 1,2). Both systems are verified against the same observations using similar hindcast sets (1970-2019 initializations for SMYLE-NOV and 1970-2017 initializations for DPLE-NOV), and skill scores are computed using consistent 20-member ensembles (see Section 2 for details). In general, the large-scale patterns of ACC skill for both seasonal temperature and precipitation, and their evolution with lead time, are very similar between these systems. However, SMYLE-NOV exhibits notably higher regional skill than DPLE-NOV, along with more widespread regions showing significantly positive ACC, particularly at short lead times. A more detailed skill comparison based on maps of surface temperature ACC difference confirms that SMYLE-NOV outperforms DPLE-NOV for lead times out to 16 months, based on the percentage of global surface area showing a significant increase in ACC (Fig. B3). There is widespread skill increase at short leads that shows

largest amplitude over South Asia and the north-eastern Indian Ocean, but there is also significant skill degradation over the North Atlantic, Greenland, and Eurasia. At longer leads (lead month 7 and beyond), the comparison grows increasingly

heterogeneous from region to region, but the areas associated with significant skill increase/decrease become roughly equal only at lead month 19 (Fig. B3). For precipitation, there is also evidence of skill improvement in SMYLE-NOV compared to DPLE-NOV at leads up to 4 months, but the global statistics indicate a rough equivalence of the two systems by lead month 7 (Fig. B4). Whether these skill improvements derive from recent CESM model developments or from the other prediction system design differences that distinguish SMYLE-NOV from DPLE-NOV (see Section 2) is unclear, but the large skill

increases at short leads do suggest that the more realistic initialization of the atmosphere and land components in SMYLE is likely an important factor. This preliminary comparison is a hopeful sign that prediction system development (achieved through advances in both modelling and incorporation of observations) can expand the spatiotemporal bounds of skillful prediction. More work is needed to develop a deeper, process-level understanding of the skill patterns shown in Figures 1 and 2, the sensitivity of process representation to system design, and where and how further improvement might be possible.

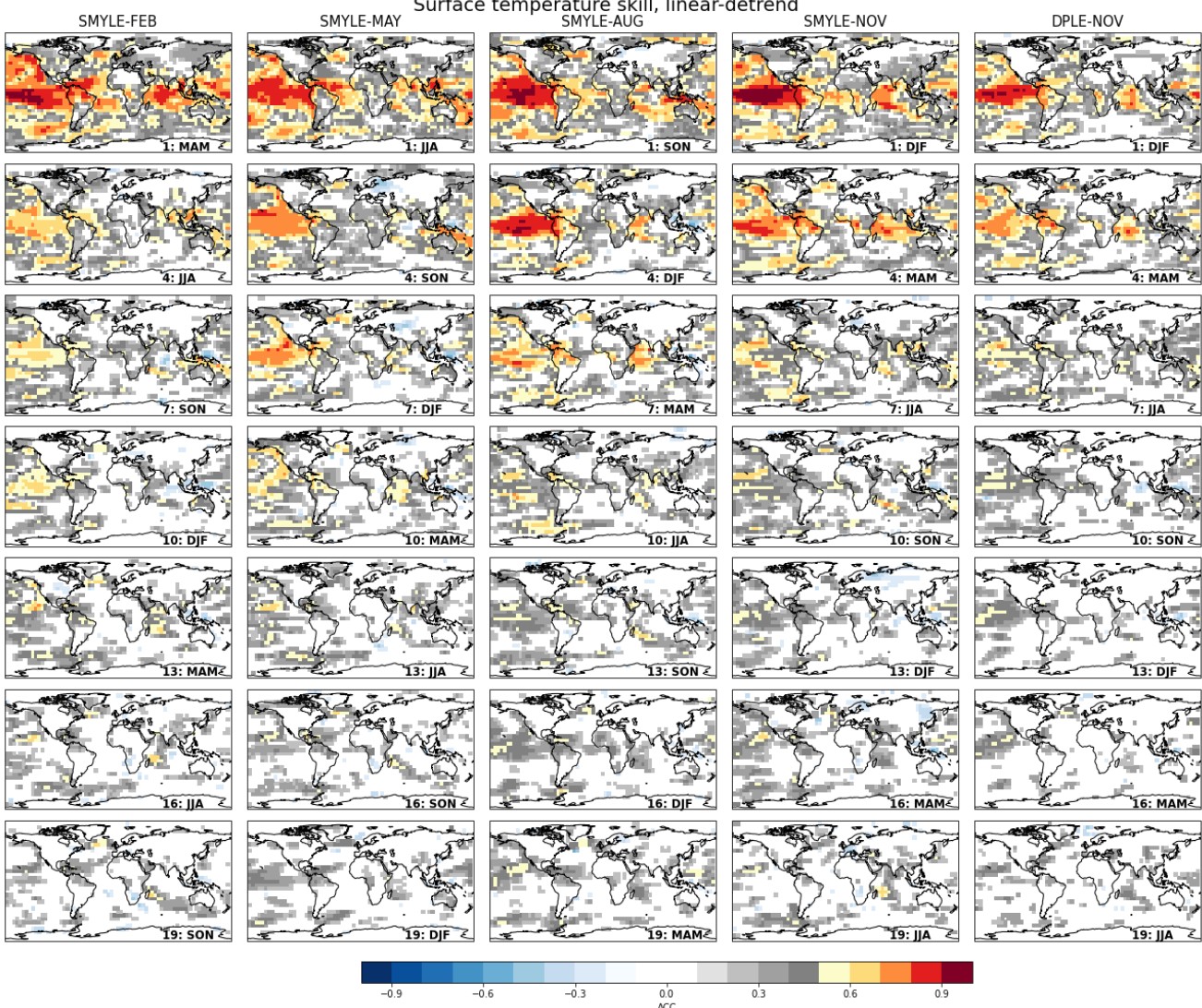

Figure 1: Anomaly correlation coefficient (ACC) for surface temperature after removing a linear trend. Columns correspond to different 20-member hindcast sets from SMYLE, with the far-right column showing 20-member DPLE results. Rows correspond to forecast season as indicated by labels that give forecast lead time and target season. Correlations are plotted only where significant (p<0.1). Verification is against a blend of CRU-TS4.05 (over land) and HadISST1 (over ocean) that spans 1970-2020. Figure B1 shows corresponding maps for non-detrended data and Figure B3 shows where SMYLE-NOV differs significantly from DPLE-NOV.

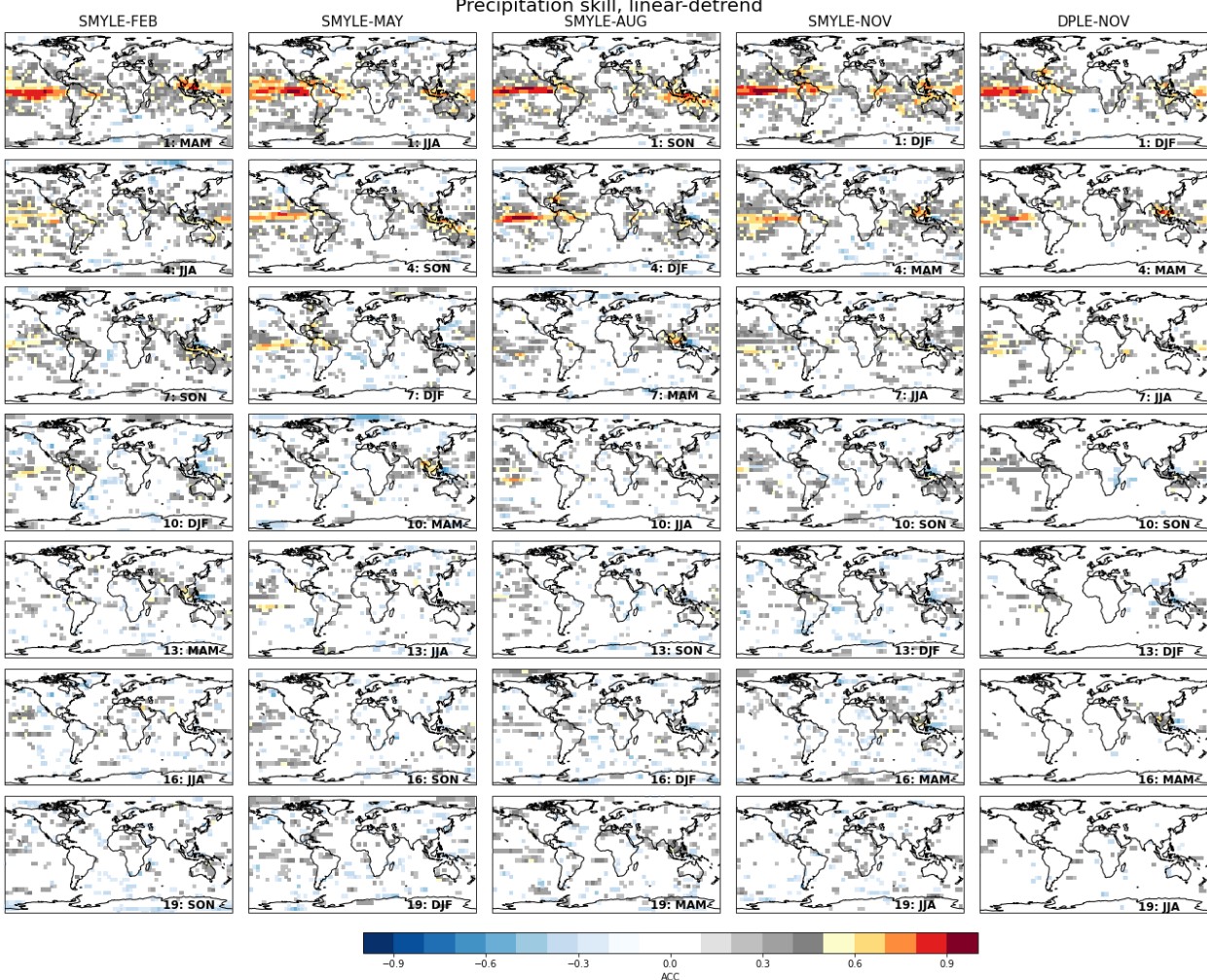

Figure 2: Anomaly correlation coefficient (ACC) for precipitation after removing a linear trend. Columns correspond to different 20-member hindcast sets from SMYLE, with the far-right column showing 20-member DPLE results. Rows correspond to forecast season as indicated by labels that give forecast lead time and target season. Correlations are plotted only where significant (p<0.1). Verification is against the GPCP v2.3 (Adler et al. 2018) dataset that spans 1979-2021. Figure B2 shows corresponding maps for non-detrended data and Figure B4 shows where SMYLE-NOV differs significantly from DPLE-NOV.

## 3.2 Sea Surface Temperature

Variations in tropical SST provide the essential underpinnings for climate prediction on seasonal to interannual timescales (Palmer and Anderson 1994; Troccoli 2010; Merryfield et al. 2020). ENSO is a dominant driver of global climate variability on interannual timescales and is therefore an important consideration in the performance appraisal of multiyear prediction systems like SMYLE. In this section, we first compare the ENSO skill from SMYLE with that obtained in other prominent

seasonal prediction systems, and then evaluate SMYLE performance at predicting seasonal anomalies in the tropical Atlantic and Indian Oceans.

### 3.2.1 El Niño-Southern Oscillation (ENSO)

The persistently high ACC for SST over the tropical Pacific in Fig. 1 is associated with long-lead predictability of ENSO events. The ENSO prediction skill in SMYLE is assessed based on SST anomalies averaged over the Niño-3.4 region (5°S-5°N, 170°-120°W; Fig. 3). The ACC for Niño-3.4 for SMYLE forecasts initialized from 1970-2019 remains above 0.5 out to lead month 11 when averaged across the four initialization months (black curve; Fig. 3i). This high mean skill extends out lead month 14 when the forecasts are subsampled to only include 1982-2016 initializations (blue curve; Fig. 3i). Skill for boreal winter (DJF) Niño-3.4 is high (ACC~0.6) even when hindcasts are initialized in February (10-month lead; Fig. 3a), and there is evidence of potentially useful (ACC~0.5) multiyear skill (e.g., the second DJF season from SMYLE-AUG corresponding to lead month 16; Fig. 3e). Forecast error is quite stable from summer through winter but grows rapidly during boreal spring (Fig. 3b,d,f,h). SMYLE-MAY has the largest error for lead month 1 seasonal hindcasts (Fig. 3d), while SMYLE-NOV shows the smallest error (Fig. 3h). The nRMSE metric yields 1 for a climatology forecasts (Section 2), and so SMYLE error for seasonal Niño-3.4 can beat climatology up to lead month 19 for target seasons in boreal fall and winter (SMYLE-FEB and SMYLE-MAY; Fig. 5b,d).

SMYLE compares well with an 8-model set of 12-month Niño-3.4 hindcasts spanning 1982-2016 from the North America Multimodel Ensemble system (NMME; Barnston et al. 2019). Slightly higher ACC and smaller RMSE are obtained using a 1982-2016 forecast set compared to a 1970-2019 forecast set, suggesting that ENSO prediction skill may be time-dependent (Fig. 3, compare black and blue curves). For lead times up to 11 months, SMYLE skill very closely matches that of the NMME multimodel mean (MMM) when averaged over the four initialization months (Fig. 3i). SMYLE skill exceeds that of all NMME models (as well as the MMM) for February-initialized forecasts (Fig. 3a,b), and it shows a very close match to the NMME MMM for November-initialized forecasts (Fig. 3g,h). The Niño-3.4 prediction skill of SMYLE is also very comparable to other operational seasonal forecast systems, such as the ECMWF seasonal forecast system 5 (SEAS5), which shows an ACC of ~0.6 at 12-month lead time during 1981-2016 when averaged over the same four initialization months as in SMYLE (Johnson et al. 2019). The corresponding ACC value from SMYLE is also ~0.6 (Fig. 3i, blue curve at lead time 12). The ENSO skill from these dynamical prediction systems is lower than that obtained from statistical prediction systems based on machine learning methods, such as the Ham et al. (2019) system in which the all-season ACC for 1984-2017 remains above 0.5 for 17 months. The lower skill of dynamical forecast systems is likely related to inherent model bias and initialization errors.

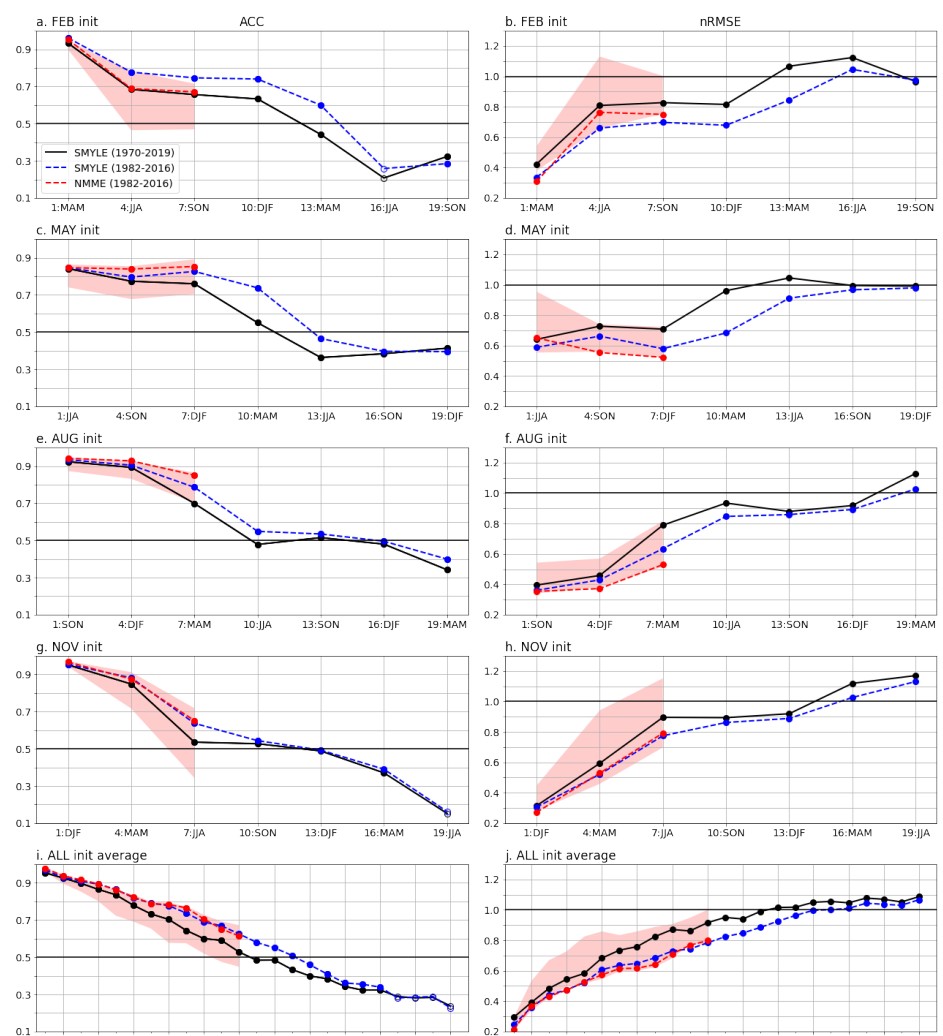

Figure 3: (left) Anomaly correlation coefficient (ACC) and (right) normalized RMSE skill scores for Niño-3.4 SST (regional average over 170°W-120°W, 5°S-5°N). Panels (a)-(h) show skill for seasonal mean Niño-3.4 for 4 different start months (FEB, MAY, AUG, NOV), with x-axis showing lead time in months and target season. Panels (i) and (j) show skill for monthly mean Niño-3.4 averaged across all 4 start months as a function of lead time in months. Filled symbols for ACC (left panels) indicate significant correlation scores (p<0.1). SMYLE skill is shown for two different forecast sets: the full set initialized 1970-2019 (solid black curve) and a subset initialized 1982-2016 (dashed blue curve). The latter can be directly compared to NMME results, which are shown as scores for the full multimodel ensemble mean (dashed red curve) as well as the range of scores from 8 individual NMME models (red shading). Verification dataset is HadISST1.

Figure 4 compares the time series of the seasonal (DJF) Niño-3.4 index in observations and SMYLE hindcasts with lead times ranging from 1 to 19 months. The NOV (1-month lead) and AUG (4-month lead) hindcasts show very high ACC (~0.9 or better), low nRMSE, and small ensemble spread (Fig. 4a,c). Skill progressively degrades in the MAY and FEB hindcast sets (Fig. 4e,g), but the latter still yields potentially actionable skill (ACC ~0.6) at a lead time of 10 months. The skill of

SMYLE-FEB is associated with good predictions of strong winter El Niño (e.g., 1973, 1983, 1998, and 2016) and La Niña (e.g., 1974, 1989, 1999, and 2000) events. This indicates that ENSO forecasts initialized in February can overcome the spring predictability barrier for strong events, possibly due to the initial large upper ocean heat content anomalies in the
equatorial Pacific (Meinen and McPhaden 2000; McPhaden 2003). SMYLE ACC skill ranges from 0.49 to 0.41 for lead times exceeding one year (Fig. 4, right column), but there are indications that certain years or decades are much more predictable than others at long lead times. For example, the multiyear La Niña events that followed the strong El Niños in 1983 and 1998 are well-captured in the 19-month lead forecasts, but the multiyear La Niña event that followed the strong El Niño of 2016 is not well predicted. In the 1980s, the AUG and MAY hindcasts (16- and 19-month lead, respectively) show
an excellent match with observations and smaller ensemble spread than for other decades. The explanation for such interannual and decadal variations in ENSO predictability and SMYLE hindcast performance will be the topic of future research.

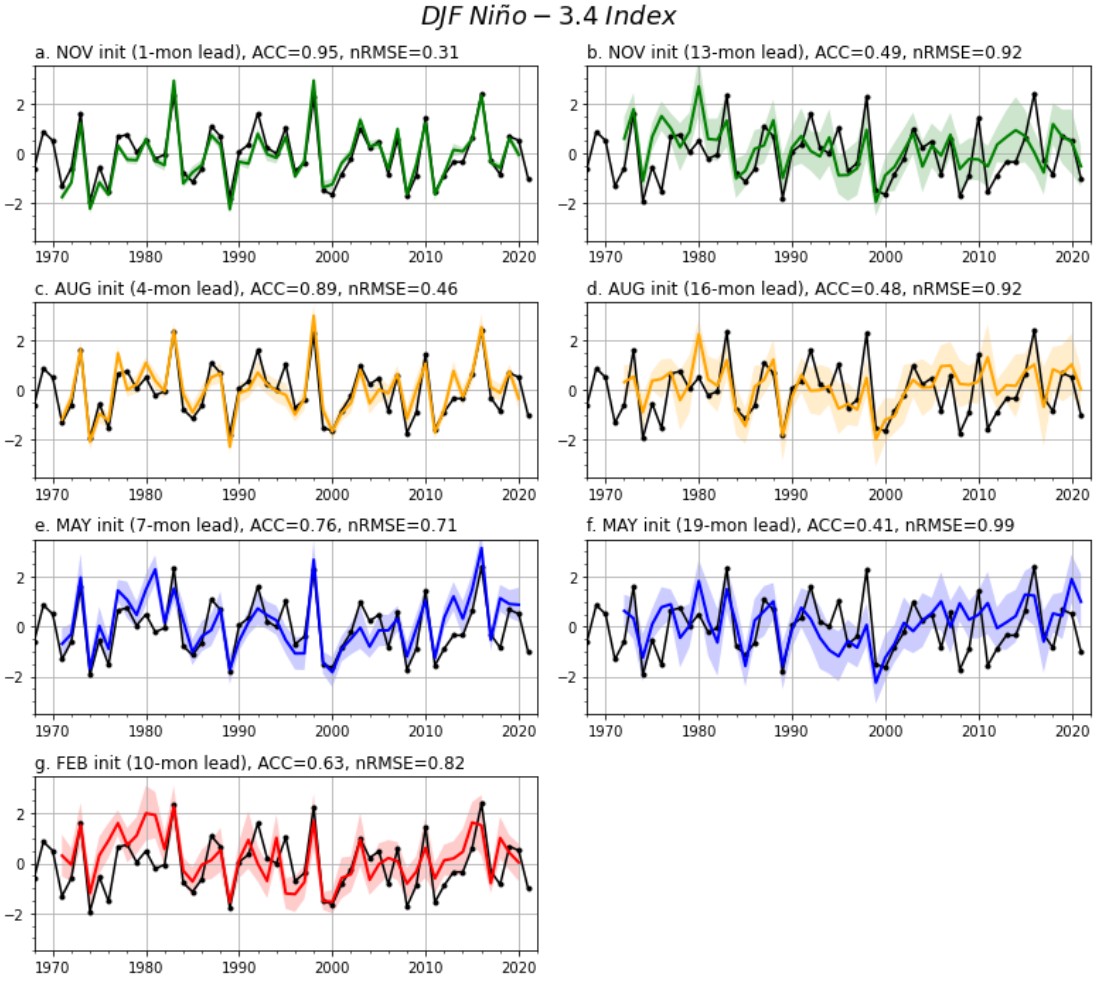

Figure 4: Time series of DJF Niño-3.4 index in observations (black curve; HadISST1) and SMYLE hindcasts at 7 different lead times.
Colored curves show the SMYLE ensemble mean and shading shows ensemble spread (±1 standard deviation). The ACC and normalized
RMSE scores are given at the top of each panel. The x-axis reflects the year of January (e.g., DJF 2020 represents the average for Dec
2019, Jan 2020, and Feb 2020).

### 3.2.2 Tropical Atlantic and Indian Oceans

In addition to ENSO, interannual SST variations in the tropical Atlantic and Indian Oceans are also important for regional
and global climate variability and predictability, not least through the key role they play in tropical interbasin interactions
[see Cai et al. (2019) for a review]. Here we evaluate the seasonal-to-multiyear prediction skill for three major indices of
climate variability in the tropical Atlantic and Indian Oceans.

The dominant mode of interannual SST variability in the equatorial Atlantic region is the Atlantic Niño (Merle et al. 1980)
which is characterized by irregular periods of warming during boreal summer (Xie and Carton 2004). While the amplitude of
the Atlantic Niño is weaker than that of ENSO, and its duration shorter (spanning approximately May-August), the Atlantic
Niño shares many commonalities with its Pacific counterpart. It is a coupled air-sea mode that exhibits numerous regional
and global teleconnections, including rainfall variability across West Africa and an intensified summer monsoon over
northeast India (Lübbecke et al. 2018; Sahoo and Yadav 2021). The Atlantic Niño has also been linked to Pacific variability,
with the Atlantic Niño state in boreal summer potentially modulating the development and amplitude of ENSO (Ding et al.
2012; Ham et al. 2013, Keenlyside et al. 2013). Coupled global climate models struggle with skillful representation of the
Atlantic Niño, which is partly attributed to model bias that prevents the development of a cold tongue across the eastern
equatorial Atlantic (Nnamchi et al. 2015; Dippe et al. 2018).

SMYLE ACC scores for Atlantic Niño exhibit a strong dependence on initialization month and verification season (Fig. 5a).
Skill for boreal summer (JJA) is of most interest, as Atlantic Niño interannual variability is strongly peaked in that season
(Lübbecke et al. 2018). As expected, SMYLE-MAY yields the highest skill for JJA at lead month 1 (ACC~0.65), but
SMYLE-FEB shows low-level skill at lead month 4 (ACC>0.4, nRMSE<1). The February initialization result is in line with
the multisystem assessment of Richter et al. (2018; their Figure 1) as well as the Norwegian climate prediction system
assessment of Counillon et al. (2021; their Figure 4), although the comparisons are complicated by differences in temporal
averaging (monthly vs. seasonal). Skill for boreal spring (MAM) shows the most rapid degradation with lead time with
insignificant ACC scores for all leads greater than 1 month and largest increase in nRMSE (Fig. 5d). In contrast, boreal
winter (DJF) ACC scores remain significant (albeit low) out to 16-month leads, and skill actually increases with lead time
for boreal fall (SON) with an ACC maximum (nRMSE minimum) seen at lead month 10. There is also a rebound in boreal
winter (DJF) skill at lead month 13 (Fig. 5a, black diamond). This skill rebound in winter and fall at longer lead times may
be related to the seasonality of Atlantic Niño or it could be an artifact of potentially correctable initialization shocks in the

prediction system. Overall, the Atlantic Niño results highlight a strong sensitivity to initialization month and target season,
and they offer an interesting contrast to the ENSO results (Fig. 3) which show a steady decline in skill after boreal spring
without any rebound.

Assessment of SST skill for the tropical Atlantic basin main development region (MDR) is important because of the region's
potential impact on tropical cyclone development during the Atlantic hurricane season (June through November). The MDR
encompasses a subtropical region extending from the Caribbean Sea to the West coast of Africa. ACC is generally greater
than 0.5 for Atlantic MDR SST for lead times up to 7 months (Fig. 5b). In contrast to Atlantic Niño, ACC and nRMSE
scores for MDR are relatively insensitive to target season, and so there is much less variability in skill with lead time (Fig.
5b,e). This could be related to the fact that MDR is a larger region, but likely also reflects the fundamentally different
dynamics at play in these two regions (Dunstone et al. 2011). At long leads, there is a curious increase in skill for May and
August initializations, and in particular, SMYLE-AUG shows a rather high ACC (~0.5) at lead month 16 (DJF target season)
which might be related to the long-range predictability of ENSO (Fig. 3c). This skill resurgence is evident to a lesser extent
in the lead month 13 of SMYLE-AUG (SON target season), which is more relevant for predicting Atlantic tropical cyclone
activity in boreal fall.

The Indian Ocean dipole (IOD), a zonal oscillation in the tropical Indian Ocean, is highly relevant for seasonal to interannual
prediction due to its teleconnections to regional climate over Africa, India, and Australia, as well as its influence on ENSO
(Saji et al. 1999; Ashok et al. 2001). ACC for IOD maximizes in boreal fall (target season SON) for all initializations (Fig.
5c), remaining above 0.5 at lead month 4 and significantly positive out to lead month 10. The ability to predict IOD in SON
could be important for anticipating winter rainfall anomalies over Australia (Ashok et al. 2003), and there is indeed some
evidence of skillful prediction of SON precipitation over the Maritime Continent and northern Australia at short leads (Fig.
2). However, the nRMSE for IOD is large even at lead month 1 (compared to the other SST indices examined), and the
errors grow with lead time while exhibiting a large sensitivity to target season (Fig. 5f). Note that the CESM2 is
characterized by a too westward extension of the eastern pole of IOD (not shown), and this likely contributes to the large
RMSE in predicting the amplitude of the IOD index.

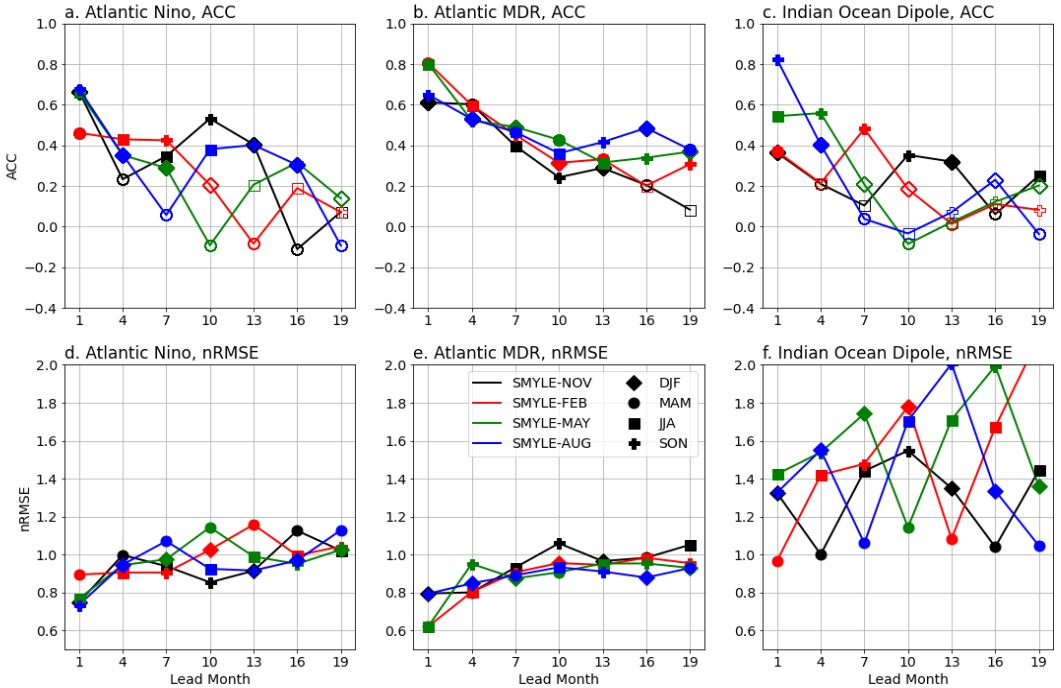

Figure 5: Anomaly correlation (top row) and normalized RMSE (bottom row) skill scores for regionally-averaged seasonal SST indices as follows: (a,d) Atlantic-Niño (20°W-0°W, 3°S-3°N); (b,e) Atlantic Main Development Region (MDR; 80°W-10°W, 10°N-20°N); (c,f) Indian Ocean Dipole (50°E-70°E, 10°S-10°N minus 90°E-110°E, 10°S-0°N). Scores for individual SMYLE start months (colors) are shown as a function of lead month, with verification season indicated by symbols as shown in the legend. Filled symbols for ACC indicate significant correlation scores (p<0.1). Time series were detrended prior to skill score computation. Verification dataset is HadISST1.

### 3.3 Sea Level Pressure

SMYLE skill at predicting seasonal variations in the large-scale atmospheric circulation is revealed by maps of ACC for sea level pressure (SLP; Fig. 6). Related to the SST prediction skill (Fig. 1), regions of useful skill (ACC>0.5) are found in the Tropics where ocean-atmosphere interaction is strong for lead times extending beyond 1 year, particularly in the eastern Pacific and over the Maritime Continent extending into the eastern Indian Ocean. Low but significantly positive SLP skill is evident even at 19-month lead time in select low-latitude regions as well as some areas in the Southern Ocean. As for temperature (Fig. 1) and precipitation (Fig. 2), SLP skill over land is generally lower than skill over ocean, and seasonal variations in the extratropics are poorly predicted even at short lead times. The modest skill at predicting SLP variations in tropical regions adjacent to land at leads greater than 12 months suggests that predictable atmospheric dynamics may be contributing to land surface temperature skill in certain low-latitude regions (cf. Figs. 1,6). As was also observed for precipitation, SLP skill exhibits low sensitivity to the removal of a linear trend (cf. Figs. 6, B5). Comparing SLP skill from SMYLE-NOV to that from DPLE-NOV reveals improved skill in the former for all leads (Fig. 6, rightmost two columns).

Maps of significant SLP skill difference between the two systems confirm this visual impression, but also highlight areas of skill degradation in SMYLE that change with lead time but appear most consistently in the pan-Atlantic region (Fig. B6). There are suggestions of a possible connection between areas of skill degradation in surface temperature and SLP (cf. B3, B6), particularly at early leads when SMYLE temperature skill over the North Atlantic is significantly lower than in DPLE.

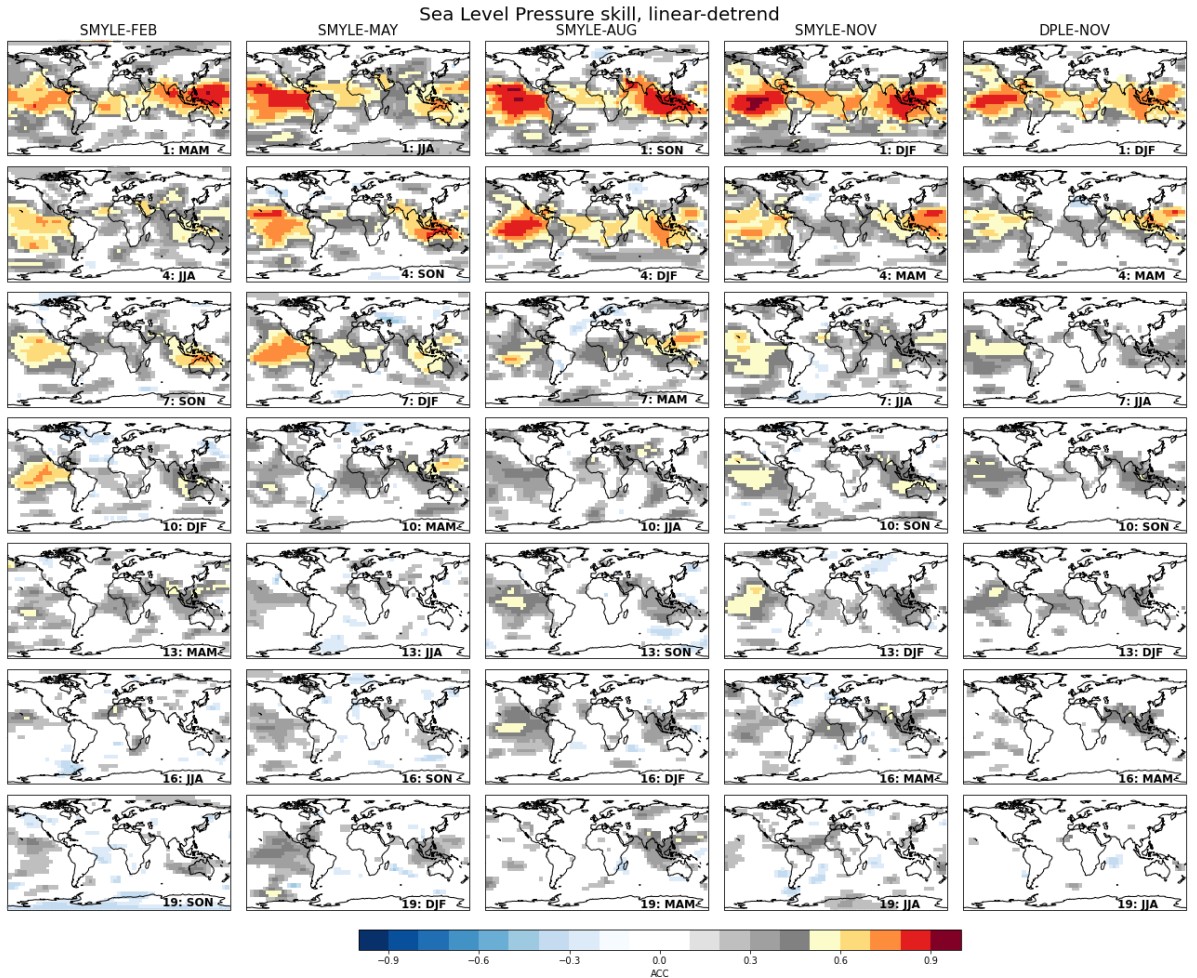

Figure 6: Anomaly correlation coefficient (ACC) for sea level pressure after removing a linear trend. Columns correspond to different 20-member hindcast sets from SMYLE, with the far right column showing 20-member DPLE results. Rows correspond to forecast season as indicated by labels that give forecast lead time and target season. Correlations are plotted only where significant (p<0.1). Verification dataset is ERA5 reanalysis. Figure B5 shows corresponding maps for non-detrended data, and Figure B6 shows where SMYLE-NOV differs significantly from DPLE-NOV.

Some dynamical forecast systems have achieved noteworthy skill at predicting year-to-year variations in the winter NAO index at lead times ranging from 1 month (Scaife et al. 2014) out to even 13 months (Dunstone et al. 2016). The ability to

predict winter NAO has important implications for anticipating winter climate impacts over Europe and North America, for even if such impacts are not themselves well-predicted, combined dynamical-statistical predictions may be possible (Simpson et al. 2019). The SMYLE set of hindcasts do not show any significant skill for DJF NAO even at the shortest lead time of 1 month, regardless of how the NAO index is computed (station-based or EOF-based; Fig. 7a,b). Furthermore, there is no evidence of useful seasonal-to-interannual ACC skill for NAO in any target season from SMYLE, and nRMSE scores are close to or above 1 at all lead times (Fig. 7). Skill from 40-member DPLE-NOV is slightly better than 20-member SMYLE-NOV (higher ACC and lower nRMSE), but the station-based NAO skill confidence interval for 20-member DPLE-NOV is too wide to definitively conclude that SMYLE skill is worse than DPLE. SMYLE-NOV skill does appear to be significantly worse than DPLE-NOV at lead month 7, and significantly better at lead month 19 where SMYLE-NOV yields a significantly positive ACC, in line with SLP skill differences in the North Atlantic sector (Fig. B6). However, obtaining such significant values through chance is not out of the question. NAO prediction skill can vary considerably with verification window (Shi et al. 2015), and there is evidence that predictability increased in the 1980s and 1990s (Weisheimer et al. 2017). Indeed, the DPLE-NOV skill for DJF NAO becomes significant for lead month 1 (ACC~0.45) if the forecast initializations are subsampled to only include 1981-2015 (to roughly match the verification window used in Dunstone et al. 2016), while the corresponding SMYLE-NOV score remains low (and now appears significantly lower than DPLE-NOV; Fig. B7). The lack of NAO skill in SMYLE (and apparent degradation relative to the DPLE system) is a clear target for future CESM prediction system improvement. While it is not clear which specific design features of the Met Office forecast systems (GloSea5, Scaife et al. 2014; DePreSys3, Dunstone et al. 2016) account for high NAO skill, noteworthy differences from SMYLE include the use of higher horizontal resolution (60km in atmosphere; 25km in ocean) and a high-top atmospheric model that can simulate stratospheric processes including the Quasi-Biennial Oscillation. Work is underway to test whether higher vertical resolution in the atmosphere and a better-represented stratosphere yields higher NAO skill relative to these baseline SMYLE results, although a robust connection between atmospheric vertical resolution and NAO skill has not been demonstrated (Butler et al. 2016). Finally, we note that lack of seasonal NAO skill does not necessarily imply a lack of skill for longer period NAO variability. The DPLE system has been shown to exhibit high skill at predicting decadal fluctuations in winter NAO (Athanasiadis et al. 2020), despite showing rather low skill for seasonal NAO (Fig. 7).

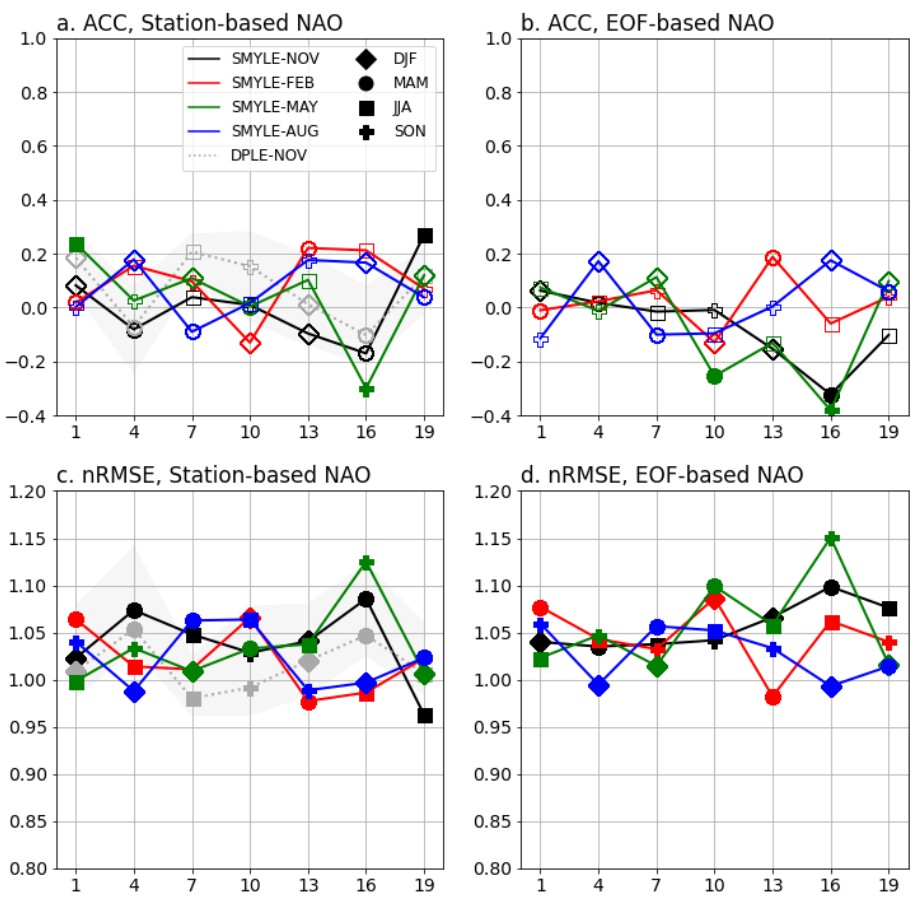

Figure 7: Anomaly correlation coefficient (ACC; top panels) and nRMSE (bottom panels) for NAO as a function of initialization month and target season. Results are shown for both station-based (left; computed as the normalized SLP difference between model grid cells nearest to Lisbon, Portugal and Reykjavik, Iceland) and EOF-based (right; computed as the PC time series of the first EOF of SLP in the domain 90°W-40°E, 20°N-80°N) seasonal NAO. Filled symbols in (a,b) indicate significant ACC scores (p<0.1). Scores from 40-member
DPLE-NOV are shown in dashed grey, with grey shading giving the 90% confidence bounds obtained from 100 resamplings (across members) of 20-member DPLE-NOV. Verification dataset is ERA5.

### 3.4 Ocean Biogeochemistry

The inclusion in SMYLE of initialized, prognostic ocean BGC fields via the MARBL module of CESM2 (Long et al. 2021) permits exploration of the predictability of marine ecosystems (Fig. 8) and ocean carbonate chemistry (Fig. 9). In Figure 8, SMYLE skill is quantified in terms of potential predictability by evaluating hindcasts against SMYLE-FOSI over the longest possible verification window (1970-2019), rather than against *in situ* observations which are relatively sparse and comprise short records. In Figure 9, SMYLE is verified against an observation-based dataset that has broad spatiotemporal coverage to
give a measure of actual BGC prediction skill.

The basic elements of marine ecosystems and the biological carbon pump, such as net primary productivity (NPP), zooplankton carbon pools (Zoo C), and carbon export to depth (C export), are well-predicted on seasonal timescales over much of the global ocean (Fig. 8). The global patterns of skill are mostly consistent across different ocean BGC fields, suggesting common sources of predictability such as regional physical drivers. C export and Zoo C, however, appear to have higher potential predictability than NPP for the Southern Ocean in the SMYLE-MAY forecasts (Fig. 8a-c). There is particular interest in developing capacity to predict near-term changes in coastal Large Marine Ecosystem (LME) regions that delineate distinct marine environments of high societal value (Krumhardt et al. 2020; black lines in Fig. 8a-c). The high predictability of BGC fields in LMEs suggests that SMYLE forecasts could potentially be used for fishery applications. A comparison of ACC skill for the California Current and Southeast US Shelf LMEs (Fig. 8 panels d-f and g-i, respectively) reveals considerably higher and longer-lasting skill for the latter region. Correlation scores for detrended NPP, carbon export, and Zoo C remain above 0.5 for boreal summer and fall seasons even out to lead month 19 for the Southeast US LME. High skill in this region would appear to be related to the overall high predictability within the Atlantic subtropical gyre (Fig. 8 a-c), which has been linked to predictable variations in nutrient limitation (Krumhardt et al. 2020). In contrast to the Southeast US region, the California Current LME exhibits different levels of skill for different BGC components, with considerably higher ACC for carbon export than for NPP or Zoo C. Nevertheless, ACC for Zoo C remains above 0.5 for boreal fall for lead times of up to 13 months (Fig. 8f). Both LMEs shown in Figure 8 (panels d-i) exhibit higher potential predictability in ecosystem variables during the summer/fall than for winter/spring. We hypothesize that this is due to the persistence of initialized subsurface nutrient anomalies, as demonstrated in Park et al. (2019). Wintertime mixing of the water column causes the reemergence of these deep nutrient anomalies into the upper ocean and subsequently affects ecosystem productivity and export during the following growing season, leading to more skillful ecosystem predictability during these months. Further work is needed to assess the practical utility of such extended range marine ecosystem forecasts for LME management.

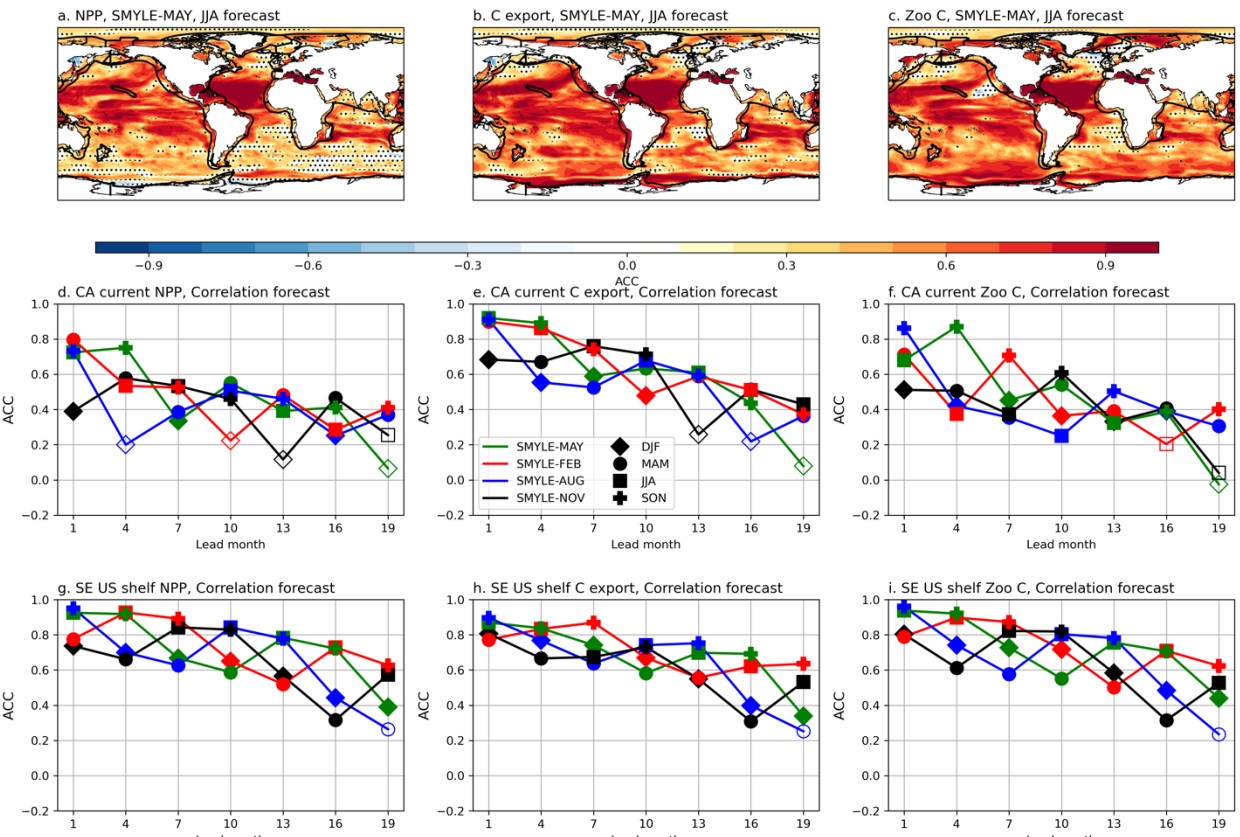

Figure 8: Anomaly correlation coefficients for SMYLE-MAY lead month 1 (JJA target season) forecasts of (a) net primary productivity, (b) carbon export, and (c) zooplankton carbon. ACC skill for spatially averaged seasonal fields as a function of lead month for (d-f) the California Current System and (g-i) the Southeast US Shelf regions. Large Marine Ecosystem (LME) boundaries are shown by black lines. Filled symbols in (d-i) indicate significant ACC scores (p<0.1).

Ocean acidification will be a steadily increasing stressor on marine ecosystems in the future as the atmospheric $CO_2$ continues to rise in response to emissions. Short-term fluctuations in pH driven by variations in circulation and tracer anomalies, however, could temporarily exacerbate the acidification problem in specific regions of interest. Figure 9 shows SMYLE skill at predicting regional anomalies in aragonite saturation state ($\Omega_{arag}$), a measure of ocean acidification. Here, SMYLE output with climatological drift removed was compared to OceanSODA-ETHZ (Gregor and Gruber 2021), a gridded observation-based data product, to determine the skill for seasonally-averaged anomalies at various lead times over the verification window 1985-2018 (determined by OceanSODA-ETHZ availability). Given the strong sensitivity of $\Omega_{arag}$ to increasing atmospheric $CO_2$, time series were first detrended to highlight the ability to predict deviations from a linear trend. A persistence benchmark forecast (computed by persisting forward in time the most recently observed seasonal anomaly at the time of SMYLE initialization) is included for comparison. Three regions were selected for analysis: the California Current LME (CA Current; see Fig. 8), the Niño3.4 region, and the subpolar North Atlantic Ocean (55-60°N, 30-

50°W). The highest skill is found for Niño3.4, where ACC remains above 0.5 out to lead month 16 for most initialization times. This implies there is a slightly longer predictability window for $\Omega_{arag}$ than for SST in this region (cf. Figs. 3, 9b). In the North Atlantic, skill scores are lower than for Niño3.4 at short lead times, but skill remains stable even out to multiyear

timescales (Fig. 9c). Skill for the CA Current is highest in boreal fall and winter seasons when initialized during the spring upwelling season (Fig. 9a; green curve corresponding to SMYLE-MAY). The May initialization also yields the highest skill for the spring upwelling season the following year (r~0.6 at lead month 10).

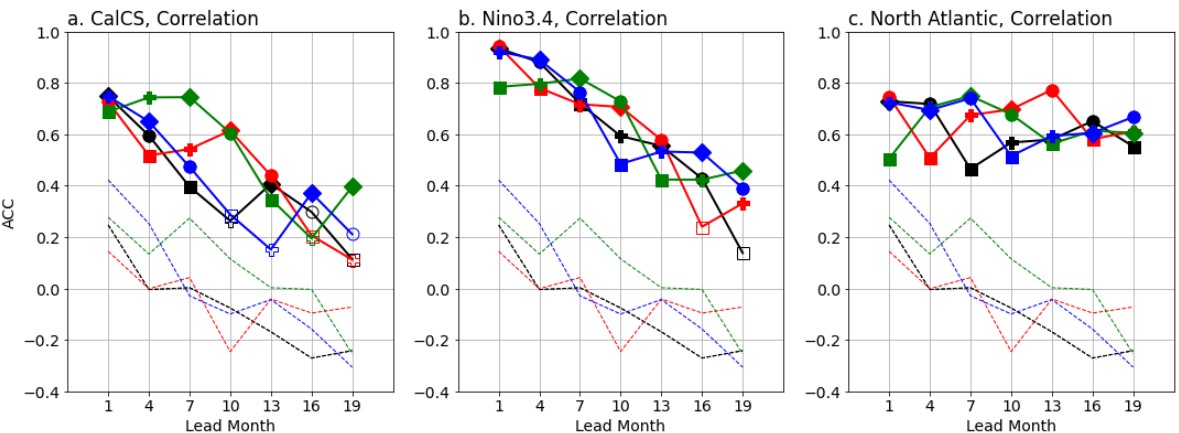

Figure 9: Anomaly correlation coefficients for aragonite saturation state ($\Omega$arag) averaged over (a) the California Current System LME, (b) the Niño3.4 region, and (c) the subpolar North Atlantic (55-60°N, 30-50°W). Colored solid lines and symbols denote SMYLE initialization month and verification season, respectively. Colored dashed lines give persistence forecast results for different initialization months. Filled symbols in (a-c) indicate significant ACC scores (p<0.1).

### 3.5 Land

Land initialization contributes to Earth system predictability across a broad range of time scales due to the inertia of climatically relevant fields such as soil moisture, vegetation, and snow cover (Merryfield et al. 2020, and references therein). In addition to adding important reservoirs of memory into the coupled system, land initialization permits exploration of the

predictability of land hydrology (Esit et al. 2021), carbon uptake (Lovenduski et al. 2020), and vegetation state (Alessandri et al. 2017). While additional sensitivity experiments will be needed to quantify the specific contribution of land initialization to prediction performance in SMYLE, initialized SMYLE hindcasts permit evaluation of land state potential predictability even when observations are sparse. Here, we demonstrate SMYLE capabilities by examining two regional case studies of land fields that exhibit high multiyear predictability: terrestrial water storage (TWS) and gross primary productivity (GPP).

The forced CLM5 simulation used to initialize SMYLE (referred to as land-only) is used as an observational proxy for forecast verification.

TWS has been the focus of several recent decadal prediction studies given its significant low-frequency variability and high relevance to water management decision-making on multiyear to decadal time horizons (Yuan and Zhu 2018; Zhu et al. 2019; Jensen et al. 2020). The CLM5 model used in SMYLE has been shown to do a good job at simulating TWS (Lawrence et al. 2019). SMYLE exhibits significant long-lead skill for TWS in select regions, with the US Southwest standing out as a location of particularly high potential predictability on multiyear time scales (Fig. 10a; the JJA season is shown, but other seasons give qualitatively similar maps). TWS in the US Southwest is dominated by decadal variability, and this region has been in extended drought conditions since the turn of the century (Fig. 10b), likely due to the combined effects of internal and anthropogenically-forced variability (Lehner et al 2018, Williams et al. 2020). The predominance of low-frequency variability implies that persistence of (initialized) decadal anomalies is an important contributor to skill in this region, and the CESM2-LE ensemble average implies that external forcing has played a role in the recent downward trend in Southwest TWS (Fig. 10b). Initialized SMYLE hindcasts outperform persistence forecasts out to about 16-month lead times, and they are more skillful than the uninitialized CESM2-LE projection at all lead times (Fig. 10c,d). Here, persistence skill is computed from the lag autocorrelation of detrended JJA TWS anomalies in the land-only simulation that was used to initialize SMYLE. The lag-1 autocorrelation corresponds to a prediction lead time of 9 months, and the lag-2 autocorrelation corresponds to a lead time of 21 months. These two persistence scores are used as benchmarks for SMYLE hindcasts at lead times less than and greater than one year, respectively. The improvement relative to CESM2-LE clearly demonstrates the value of initialization in addition to external forcing for predicting regional climate on multiyear timescales. The improvement relative to persistence suggests that there is some skill gain through prediction of TWS drivers in SMYLE, particularly in the first year (e.g., skillful prediction of seasonal precipitation over this region at short lead times; see Fig. 2).

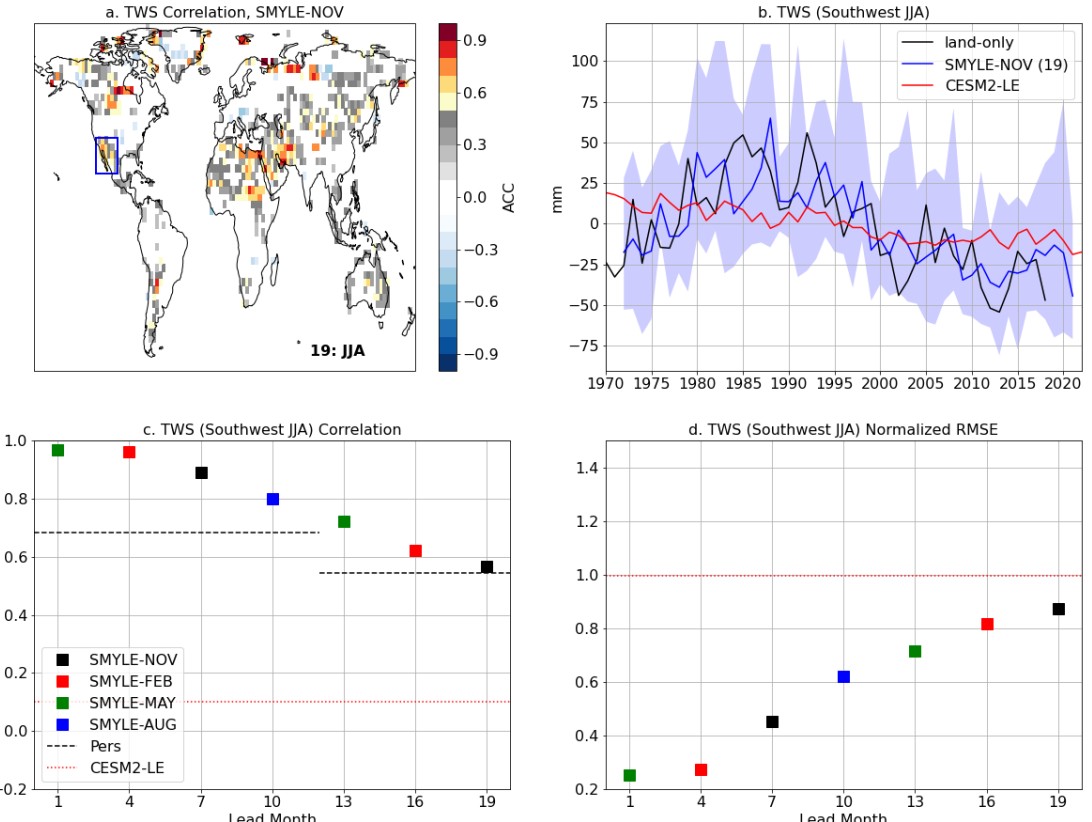

Figure 10: (a) ACC for terrestrial water storage (TWS) in boreal summer (JJA) at lead month 19 (from SMYLE-NOV). The blue box (120°W-100°W, 22°N-37°N) defines a "Southwest" region. (b) Time series of raw Southwest JJA TWS (in units of mm; anomalies from 1972-2018 climatology) from land-only simulation (black), SMYLE-NOV (blue; lead month 19 with ensemble mean/range given by line/shading), and CESM2-LE (red; 50-member mean). (c,d) ACC and nRMSE scores for JJA TWS over the Southwest region. Time series were detrended prior to skill score analyses shown in panels a,c,d, but panel b shows the non-detrended data. Dashed lines in (c) show skill for persistence forecasts computed from the lag autocorrelation of detrended JJA TWS from the land-only simulation, and red dotted lines in (c,d) give the CESM2-LE scores (which are independent of lead month).

GPP is another land field that exhibits noteworthy multiyear potential predictability in select regions such as Siberia (Fig. 11a). GPP variability over Siberia is characterized by large multiyear fluctuations superimposed on a significant externally-forced upward trend (Fig. 11b). Skill maps for non-detrended seasonal GPP show widespread regions of high skill on every continent, with much of that skill presumably associated with external forcing (not shown). The skill scores obtained after detrending (Fig. 11a,c,d) help to highlight the predictability conferred by initialization. ACC remains above 0.5 for detrended Siberia GPP in boreal summer out to lead month 10, and it beats persistence at all lead times (Fig. 11c). While the uninitialized CESM2-LE ensemble accurately captures the upward trend in GPP over Siberia, SMYLE greatly improves the skill at predicting near-term deviations from the linear trend (Fig. 11b,c,d). This result suggests that potentially useful GPP forecasts may be possible at lead times of a year or more, but further work is needed to identify the predictability

mechanisms to bolster confidence in their use. The significant ACC scores obtained over Siberia for JJA TWS (Fig. 10a) offer a hint that high GPP skill may be related to accurate prediction of soil water availability in this region and season.

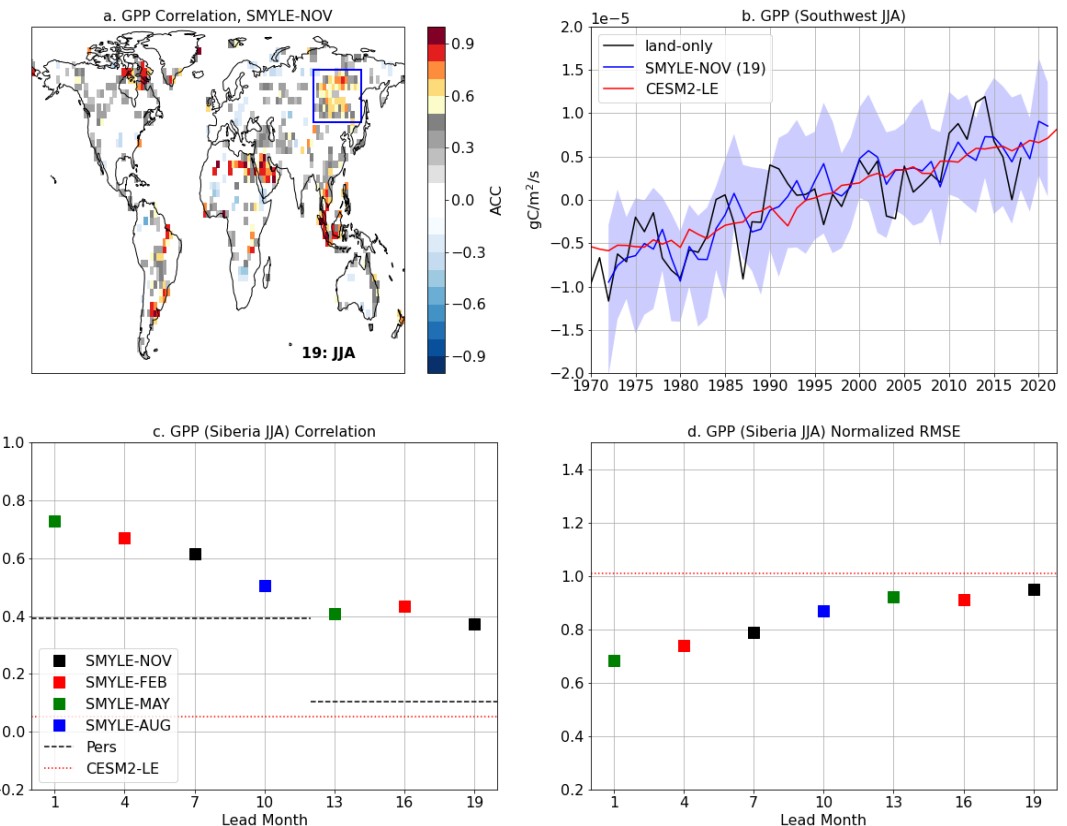

Figure 11: (a) ACC map for gross primary productivity (GPP) in boreal summer (JJA) at lead month 19 (from SMYLE-NOV). The blue box (92°E-137°E, 45°N-67°N) defines a "Siberia" region. (b) Time series of raw Siberia JJA GPP (in units of gC/m2/s; anomalies from 1972-2018 climatology) from land-only simulation (black), SMYLE-NOV (blue; lead month 19 with SMYLE ensemble mean/range given by line/shading), and CESM2-LE (red; 50-member mean). (c,d) ACC and normalized RMSE scores for JJA GPP over the Siberia region. Time series were detrended prior to skill score analyses shown in panels a,c,d, but panel b shows the non-detrended data. Dashed lines in (c) show persistence forecast scores computed as the lag autocorrelation of detrended JJA GPP from the land-only simulation, and red dotted lines in (c,d) give the CESM2-LE scores (which are independent of lead month).

## 3.6 Sea Ice

The FOSI simulation used to initialize the ocean and sea ice component models in SMYLE exhibits realistic interannual variability in pan-Arctic winter (JFM) and summer (JAS) sea ice extent (SIE), although JFM variance is somewhat lower than observed (Fetterer et al. 2017) and JAS anomalies are biased high in the decade spanning 2008-2018 (Fig. 12). Given this imperfect initialization as well as the limited observational record (from 1979 onwards), Figure 12 compares SMYLE SIE skill assessed relative to both FOSI (potential skill) and satellite observations (actual skill). In all seasons, Arctic SIE

variance is dominated by a large amplitude forced decline which can be estimated from the CESM2-LE ensemble mean (members 51-100; see Section 2). As a result, SMYLE SIE correlation scores (relative to either FOSI or observations) exceed 0.83 at all lead times for both JFM and JAS (not shown). To avoid skill score saturation associated with this forced decline, we report in Table 1 correlation and nRMSE (normalized by the standard deviation of the verification dataset) scores after removing a linear trend (although we note that removing a linear trend does not necessarily remove the forced 615 response, which is likely nonlinear--see blue curve in Fig. 12).

Detrended SIE skill is higher for JAS than for JFM at short lead times (Fig. 12; see Table 1 at lead months 2 and 5). The relatively good skill for summer SIE up to lead month 5 (Fig. 12d) appears to be related to accurate reproduction in SMYLE of the abrupt decline in summer SIE in the mid-2000s (Fig. 12d). However, the SMYLE ensemble spread fails to encompass 620 the extreme summer SIE minimum observed in 2012 even at lead month 2 (Fig. 12b). This prediction system failure may be related to biases in the initial conditions noted above, but the SMYLE spread also fails to encompass the FOSI value for JAS SIE in 2012 and the prediction error grows with lead time. The uninitialized but externally forced CESM2-LE ensemble simulates changes in the rate of JAS SIE decrease (more rapid decrease around 2000 followed by slower decrease after 2010) that yield a correlation of ~0.2 with detrended observations. This suggests that external forcing may have played a role 625 in the observed deviation from a linear decrease in summer SIE. Winter SIE exhibits lower amplitude variability about the linear trend, and this variability is dominated by interannual fluctuations. Initialization does improve the simulation of winter SIE (Fig. 12, left column; Table 1, JFM column), but the detrended skill scores are quite low even when verified against FOSI (ACC~0.37 at lead month 2). SMYLE skill scores for JFM SIE are relatively insensitive to verification dataset (satellite observations vs. FOSI), in contrast to JAS skill which is clearly higher when FOSI is used as the benchmark (Table 630 1). Overall, the SMYLE results for detrended pan-Arctic SIE appear to be in line with those reported from other seasonal prediction systems (e.g., Chevallier et al. 2013; Bushuk et al. 2017), but a dedicated multi-system study would be needed for a clean skill comparison. The present examination of pan-Arctic seasonal variability masks the considerable sensitivity of Arctic sea ice prediction skill to verification month and region (Bushuk et al. 2017), which can be a topic of future investigation with SMYLE.

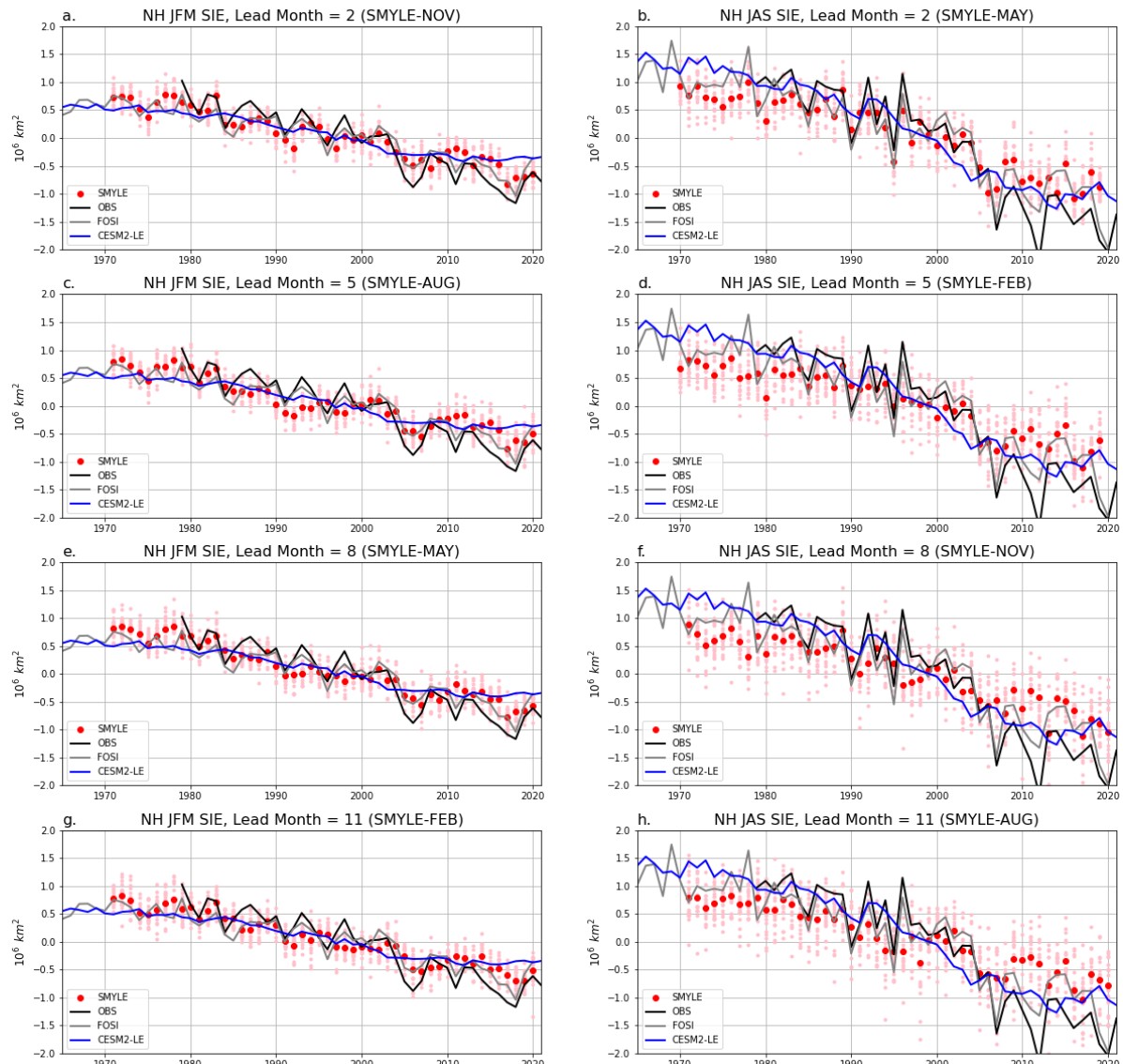

Figure 12: Northern Hemisphere seasonal sea ice extent (SIE) anomalies (relative to 1980-2015 climatology) for JFM (a,c,e,g) and JAS (b,d,f,h) from SMYLE (red and pink dots show ensemble mean and individual members, respectively), satellite observations (black line; Fetterer et al. 2017), SMYLE-FOSI (grey line), and CESM2-LE (blue line; 50-member mean). Rows show results for lead months 2, 5, 8, and 11. Skill scores (correlation and normalized RMSE) are provided for SMYLE (red) and CESM2-LE (blue) relative to both observations and SMYLE-FOSI (scores in parentheses). All time series were detrended (over the window of overlap between SMYLE and the verification dataset) prior to computing skill scores.

|  | JFM | | JAS | |
| --- | --- | --- | --- | --- |
| Data \ Verification | Obs | FOSI | Obs | FOSI |
| CESM2-LE | 0.13, 1.01 | -0.08, 1.11 | 0.16, 1.04 | 0.03, 1.11 |
| FOSI | **0.86**, 0.51 | 1, 0 | **0.85**, 0.54 | 1, 0 |
| SMYLE (LM=2) | **0.38**, 0.97 | **0.37**, 1.03 | **0.58**, 0.82 | **0.68**, 0.73 |
| SMYLE (LM=5) | 0.31, 1.03 | 0.25, 1.15 | **0.39**, 0.93 | **0.45**, 0.90 |
| SMYLE (LM=8) | **0.32**, 0.98 | **0.28**, 1.05 | -0.01, 1.11 | 0.12, 1.07 |
| SMYLE (LM=11) | 0.22, 1.03 | 0.21, 1.08 | 0.04, 1.09 | **0.25**, 0.99 |

Table 1: Skill scores for detrended Northern Hemisphere seasonal sea ice extent (SIE) anomalies (relative to 1980-2015 climatology) for JFM and JAS seasons (see Fig. 12 for corresponding non-detrended time series). Leftmost column lists the simulations being evaluated, including SMYLE at lead months 2, 5, 8, and 11. The remaining columns show ACC (left number; in bold if p<0.1) and nRMSE (right number) evaluated against satellite observations (Fetterer et al. 2017) and FOSI. Time series were detrended prior to skill score computation.

As discussed in the review by Guemas et al. (2016), sea ice volume (SIV) is much more predictable than SIE. SMYLE skill for detrended SIV remains high (ACC>0.5 and nRMSE<1.0) even out to lead month 20 for both winter and summer seasons when verified against FOSI (Fig. 13; Table 2). This potential skill derives from accurate simulation of variations in the rate of SIV change simulated in FOSI: fast decline between about 1985-1995, a rebound between about 1995-2003, and slower decline after about 2003 (Fig. 13). It is interesting to note that the conditional bias seen in SMYLE for JAS SIE (resulting in a lower decreasing trend than seen in observations or FOSI, particularly at long leads; Fig. 12), is not evident for JAS SIV even at lead month 20 (Fig. 13). The explanation for this merits further investigation, but it implies there is a compensating conditional bias in the sea ice thickness field. The CESM2-LE exhibits negligible skill at capturing the deviations from the linear trend as reconstructed in FOSI (Table 2). The CESM2-LE exhibits negligible skill at capturing these deviations from the linear trend. While SMYLE shows high potential prediction skill for SIV, the actual skill is less clear as there are no observed timeseries for pan-Arctic SIV that can be used for verification. The Pan-Arctic Ice Ocean Modelling and Assimilation System (PIOMAS; Schweiger et al. 2011) reanalysis product is a commonly used benchmark, but the detrended SIV variability from PIOMAS does not correlate with that obtained from FOSI (Table 2). As a result, SMYLE correlations with PIOMAS are generally negative (indeed, significantly negative at long lead times; Table 2) and scores relative to PIOMAS are worse than the corresponding scores for CESM2-LE (Fig. 13). The discrepancy between the FOSI and PIOMAS reconstructions of SIV, an essential component of sea ice predictability, highlights the challenges associated with generating consistent, long timescale reconstructions of the Earth system state for use in initialized dynamical prediction.

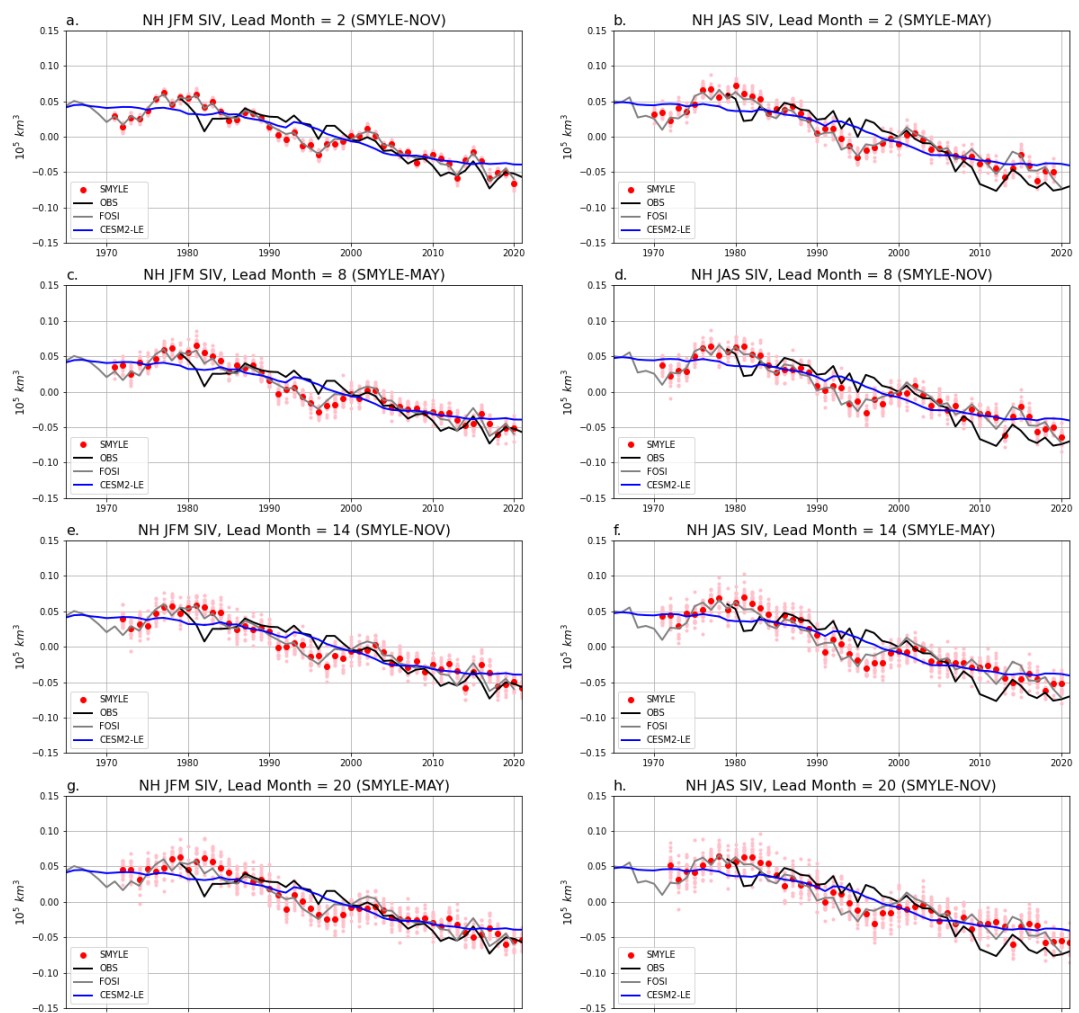

Figure 13: Northern Hemisphere seasonal sea ice volume (SIV) anomalies (relative to 1980-2015 climatology) for JFM (a,c,e,g) and JAS (b,d,f,h) from SMYLE (red and pink dots show ensemble mean and individual members, respectively), the PIOMAS reconstruction (black line; Schweiger et al. 2011), SMYLE-FOSI (grey line), and CESM2-LE (blue line; 50-member mean). Rows show results for lead months 2, 8, 14, and 20. Skill scores (correlation and normalized RMSE) are provided for SMYLE (red) and CESM2-LE (blue) relative to both PIOMAS and SMYLE-FOSI (scores in parentheses). Time series were detrended (over the window of overlap between SMYLE and the verification dataset) prior to computing skill scores.

|  | JFM | | JAS | |
|---|---|---|---|---|
| Data \ Verification | PIOMAS | FOSI | PIOMAS | FOSI |
| CESM2-LE | 0.34, 0.95 | 0.06, 1.06 | 0.09, 1.05 | -0.02, 1.09 |
| FOSI | 0.11, 1.30 | 1, 0 | 0.08, 1.33 | 1, 0 |
| SMYLE (LM=2) | -0.02, 1.47 | **0.97**, 0.27 | -0.11, 1.43 | **0.87**, 0.50 |
| SMYLE (LM=8) | -0.35, 1.66 | **0.79**, 0.63 | -0.27, 1.53 | **0.78**, 0.65 |
| SMYLE (LM=14) | **-0.45**, 1.71 | **0.67**, 0.80 | **-0.48**, 1.68 | **0.67**, 0.77 |
| SMYLE (LM=20) | **-0.55**, 1.77 | **0.57**, 0.88 | **-0.52**, 1.65 | **0.50**, 0.94 |

Table 2: Skill scores for detrended Northern Hemisphere seasonal sea ice volume (SIV) anomalies (relative to 1980-2015 climatology) for JFM and JAS seasons (see Fig. 13 for corresponding non-detrended time series). Leftmost column lists the simulations being evaluated, including SMYLE at lead months 2, 8, 14, and 20. The remaining columns show ACC (left number; in bold if $p<0.1$) and nRMSE (right number) evaluated against PIOMAS or FOSI. Time series were detrended prior to skill score computation.

### 3.7 Climate Extremes

There is great interest in developing capacity to skillfully predict shifts in the probability of high-impact weather extremes. Tropical cyclones (TCs) are prime examples of impactful weather events whose statistics are shifted by potentially predictable changes in the background climate state (e.g., Chang et al. 2020). TC activity is largely controlled by large-scale environmental conditions, including tropical SST, vertical wind shear, mid-troposphere relative humidity and low-level vorticity. Variability of these large-scale conditions, and in particular ENSO-related variability (Lin et al. 2020), can strongly modulate TC statistics in different ocean basins.

To assess SMYLE skill at predicting TC activity, we focus on interannual TC statistics and their modulation with ENSO. While individual TC events are not predictable at seasonal to interannual lead times, SMYLE exhibits promising skill at predicting some of the key environmental variables that affect year-to-year changes in TC statistics, such as tropical SST (Figs. 1,5). TCs in SMYLE are detected and tracked by applying the TempestExtremes tracking algorithm (Ullrich and Zarzycki 2017; Zarzycki et al. 2021) to the 6-hourly model output. The TC detection criteria in the tracker were adjusted to accommodate the relatively coarse model resolution. The detected global average annual number of TCs (or TC-like storms) in SMYLE lead month 1 hindcasts for the period of 1970-2018 is 69, which is less than the observed annual number of 87. However, the detected TC tracks in SMYLE exhibit realistic spatial distribution and seasonality (Figs. B8, B9).

TC activity in SMYLE is assessed during the extended summertime TC season (JJASON for the Northern Hemisphere basins and DJFMAM for the Southern Hemisphere) with a focus on basin-scale statistics. Figure 14 compares the observed regression between ENSO and global TC track density to that from SMYLE at various lead times. TC track density is defined as the total number of TCs passing through each 5˚x5˚ box during TC season, and ENSO state is quantified as the

observed Nino 3.4 index averaged over the corresponding TC season for the period of 1970-2018. Note that forecast lead time is defined as outlined in Section 2, and that the SMYLE initialization month changes with hemisphere in Figure 14. For example, the lead month 1 panel (Fig. 14b) displays results from SMYLE-MAY in the Northern Hemisphere (target season JJASON) and from SMYLE-NOV in the Southern Hemisphere (target season DJFMAM). The observed benchmark (Fig. 14a) is based on the Best Track data set that combines NOAA's National Hurricane Center (Landsea and Franklin 2013) and the U.S. Navy's Joint Typhoon Warning Center (Chu et al. 2002) products. During El Niño years, observed TC activity increases in the northwestern Pacific (NWP) and the South Pacific (SP), decreases in the North Atlantic (NA), and shifts westward in the eastern Pacific (EP). These ENSO-related variations in TC activity are well captured in SMYLE at lead times up to 16 months. However, SMYLE also shows a strong ENSO-related decrease in TC activity in the South Indian Ocean (SI) which is not seen in observations.

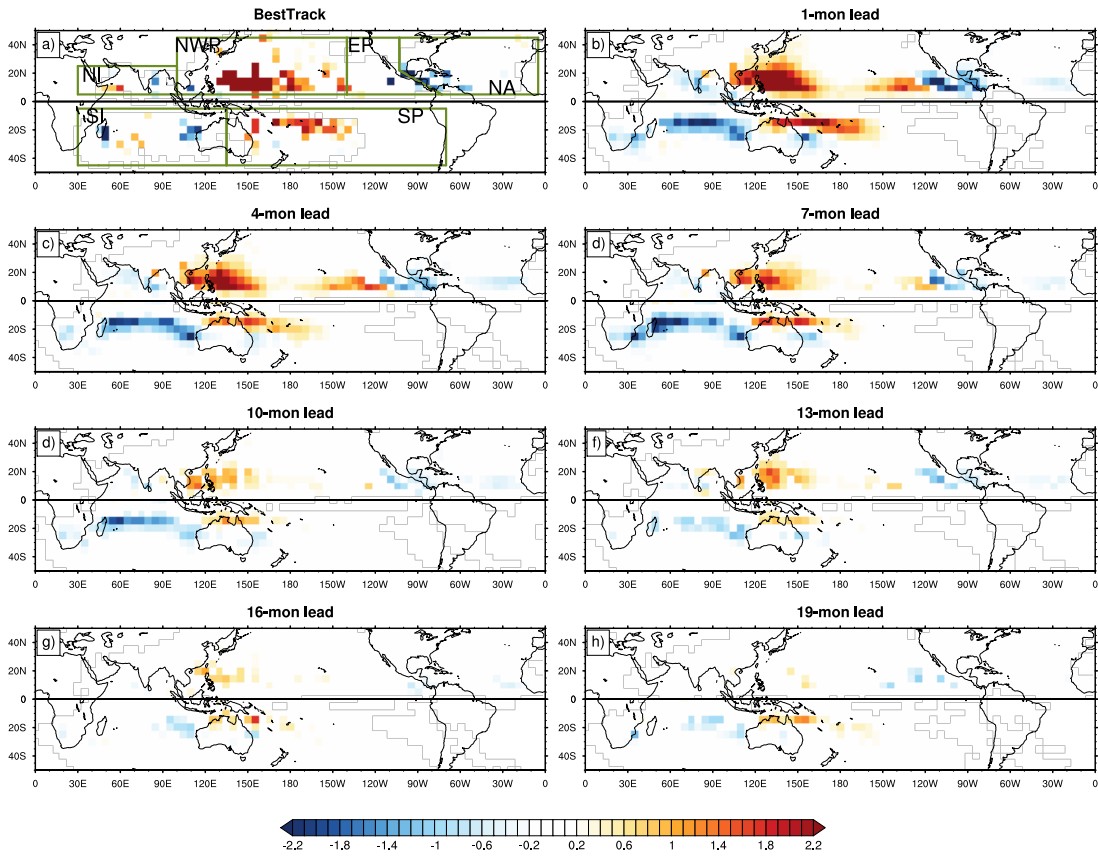


Figure 14: Regressions of seasonal-mean TC track density on the corresponding seasonal mean Nino 3.4 index (JJASON in the Northern Hemisphere and DJFMAM in the Southern Hemisphere) for the period of 1971-2018 in the (a) Best Track observations and (b-h) SMYLE forecasts. Shadings are significant at the 90% confidence level. The green lines in (a) mark the boundaries of different TC basins -- Northwestern Pacific (NWP), North Atlantic (NA), eastern Pacific (EP), North Indian Ocean (NI), South Indian Ocean (SI), and South

Pacific Ocean (SP). Note that at 19-month lead time (panel (h)), data from SMYLE-NOV is for JJASO (instead of JJASON) given the 24-month simulation length. Similarly, data from SMYLE-MAY at 19-month lead is for DJFMA (instead of DJFMAM).

Figure 15 shows the interannual variability of normalized seasonal TC accumulated cyclone energy (ACE; defined as the
sum of the squared 6-hourly maximum sustained surface wind speed (in $kt^2$) over the lifetime of a TC for all TCs within a certain basin) in NA, EP, NWP, and SP. The corresponding correlation coefficients between the normalized TC ACE from observations and SMYLE are shown in Table 3. The correlations from the NI and SI regions are not significant and are therefore not shown. The highest ACE skill is seen in the NA region, where significant positive correlations (90% confidence level) are found at all seasonal lead times apart from the 7-month lead (Nov-initialized) forecast. Skill for the
NWP region is shorter lived, but significant correlations are found even out to 13-month lead. The correlations in EP and SP are only significant out to 4-month lead. In addition to TC ACE, we also examined the interannual variability of seasonal TC number, which in general shows weaker correlations with the observations than ACE. For example, the correlations for TC numbers are significant only at 1-month and 4-month lead in NA, and only at 1-month lead in EP (Table B1). Skill for the NA and NWP regions at 1-month lead is generally comparable with other seasonal forecast models. For example, Befort et
al. (2022) evaluated TC prediction skill over the NA (during JASO) and NWP (during JJASO) for the period of 1993-2014 in six seasonal forecast systems. They found that the models on average have a correlation coefficient of 0.6 for ACE over the NA. For the NWP, the average correlation is 0.65, with 0.4 being the lowest value. Despite having a lower model resolution, SMYLE skill falls within the range of these results, although the comparison is complicated by different verification windows and different definitions of active TC season.

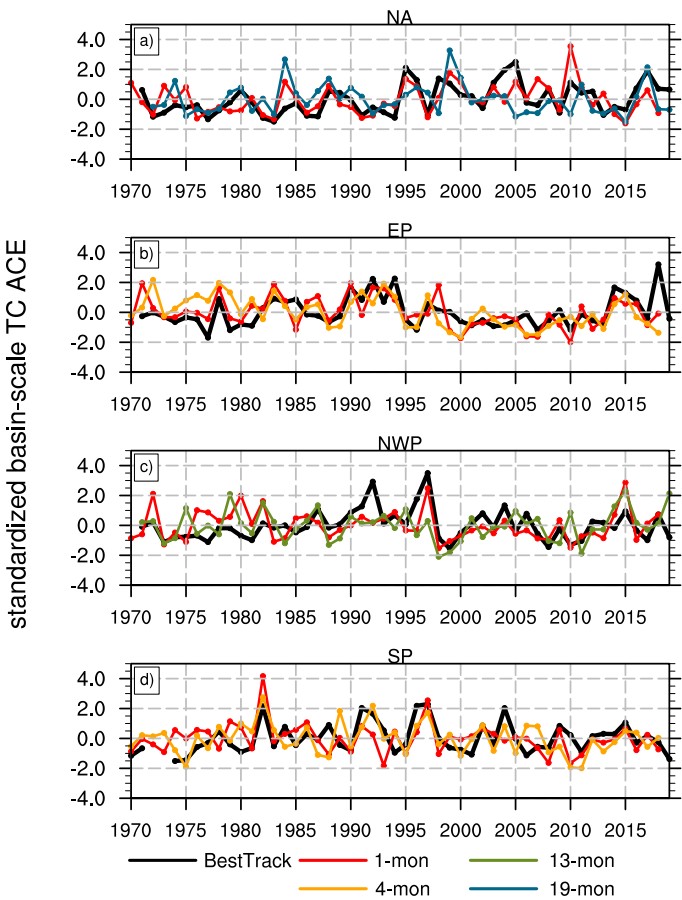

Figure 15: Time series of normalized seasonal-mean (JJASON for the Northern Hemisphere and DJFMAM for the Southern Hemisphere) TC Accumulated Cyclone Energy (ACE) for the period of 1970-2018 in the observations (black curves), SMYLE forecasts at 1-month lead (red curves), and the respective SMYLE forecasts at the longest lead that yields a significant correlation (as shown in Table 3) (colored curves) in (a) North Atlantic, (b) eastern Pacific, (c) Northwestern Pacific, and (d) South Pacific. TC basin boundaries are shown in Fig 14. Note that at 19-month lead time (panel (a)), data from SMYLE-NOV is for JJASO (instead of JJASON) given the 24-month simulation length.




| Lead | NA | EP | NWP | SP |
|---|---|---|---|---|
| 1-month | **0.55** | **0.52** | **0.44** | **0.41** |
| 4-month | **0.37** | **0.27** | **0.26** | **0.47** |
| 7-month | 0.20 | 0.22 | **0.38** | 0.12 |
| 10-month | **0.27** | -0.2 | 0.19 | 0.06 |
| 13-month | **0.28** | -0.16 | **0.24** | -0.16 |
| 16-month | **0.34** | -0.06 | 0.08 | 0.04 |
| 19-month | **0.25** | -0.02 | 0.13 | 0.12 |

Table 3: Correlation coefficients of normalized seasonal-mean (JJASON for the Northern Hemisphere; DJFMAM for the Southern Hemisphere) TC ACE between the observations and the SMYLE hindcasts (rows give leadtime). Bold numbers indicate that correlations are significant at 90% confidence level ($p<0.1$). The correlations in the North and South Indian Ocean are not significant at any leadtime and are therefore not shown. Note that at 19-month lead time (bottom row), data from SMYLE-NOV is for JJASO (instead of JJASON) given the 24-month simulation length. Similarly, data from SMYLE-MAY at 19-month lead is for DJFMA (instead of DJFMAM).

## 4 Conclusions

The SMYLE prediction system using CESM2 is a new community resource for exploring predictability of the Earth system out to a 2-year time horizon. With its relatively large ensemble size (20 members), broad temporal coverage (4 starts per year from 1970-2019), and extensive global output of fields from all component models of CESM2 (including ocean biogeochemistry), SMYLE is a large and enormously rich dataset that can facilitate rapid advancements in our understanding of seasonal to multiyear variability and predictability in the atmosphere, ocean, land, and sea ice. SMYLE represents one piece of a larger effort within CESM to move towards seamless initialized prediction for climate timescales. A subseasonal prediction system (45-day hindcasts initialized weekly) has recently been introduced that employs the same CESM2 model and nearly the same method for initializing ocean and sea ice components (Richter et al. 2022). An extension of SMYLE-NOV to decadal timescales is already underway and will be available soon. Extending SMYLE by adding missing start months (from 4 to 12 per year) would allow for more detailed studies of seasonal prediction skill, and this possibility is under consideration. Finally, the CESM2-LE future scenario simulations permit investigation of Earth system change out to 2100 (Rogers et al. 2021). As such, SMYLE occupies a heretofore neglected range within a suite of CESM2 prediction systems designed to probe possible climate futures over timescales from weeks to centuries.

The skill overview presented here, while necessarily cursory, shows that SMYLE performance is quite good albeit not groundbreaking. In terms of ENSO skill, which is of paramount importance for global prediction on these timescales, SMYLE appears to be competitive with both NMME and the SEAS5 system from ECMWF. A relative benefit of SMYLE is that the large collection of long (24-month) hindcasts permit the study of extended ENSO predictability as well as its state dependence. We have also presented evidence of promising multiyear skill (or potential skill) for marine ecosystems and

ocean carbonate chemistry, terrestrial water storage and gross primary productivity, Arctic sea ice volume, and tropical cyclone activity. A community effort will be needed for a full assessment of SMYLE forecast fidelity for various quantities of interest. This preliminary assessment has focused on deterministic skill metrics, and an evaluation of SMYLE probabilistic skill is left for future work.

The lack of skill for winter NAO stands in stark contrast to some other systems (e.g., Dunstone et al. 2016) and will require further investigation, but SMYLE represents a control dataset (as well as an experimental system) that will greatly facilitate such research. Likewise, the disappointing skill for seasonal precipitation over land, while not obviously unique to SMYLE, suggests that follow-up is needed to better understand why relatively long-lasting ENSO skill does not translate into skillful predictions of hydroclimate over land. Dunstone et al. (2020) show that interannual skill for regional monsoon precipitation

is considerably enhanced during active ENSO years, suggesting that relatively low mean precipitation skill may mask the presence of episodic forecasts of opportunity related to the state of the tropical Pacific. Work is ongoing to develop more insight into state-dependent predictability from the vast archive of SMYLE data.

The inclusion of historical state reconstructions that utilize CESM2 component models (for ocean, sea ice, land, and ocean

biogeochemistry) in the SMYLE dataset allows for assessments of potential predictability by verifying hindcasts against the reconstructions that were used for initialization. This alleviates issues associated with sparse or unreliable observations (as demonstrated above for ocean biogeochemistry, land, and sea ice) and can be very useful for detailed studies of predictability mechanisms (Yeager 2020). The choice to use JRA55 (JRA55-do) as the basis for component state reconstruction means that SMYLE could potentially be extended back in time as far as 1958, and forward in time to near real-time. We anticipate that

future SMYLE extensions will further enhance the utility of this resource.


**Appendix A**

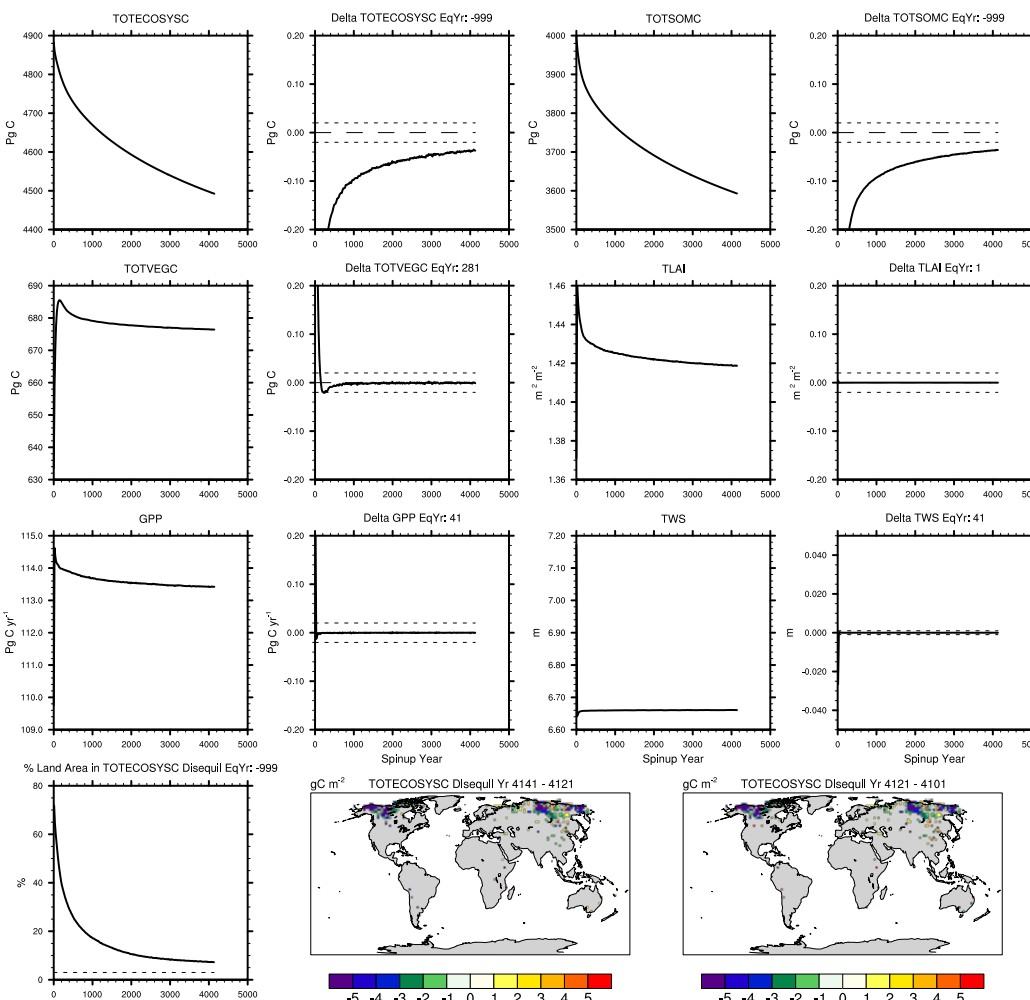

## smyle_spinup_postAD_v3 Annual Mean


Fig. A1: Evolution of land carbon fields in land-only spin-up simulation: total ecosystem carbon (TOTECOSYSC), total soil organic matter carbon (TOTSOMC), total vegetation carbon (TOTVEGC), total leaf area index (TLAI), gross primary productivity (GPP), and terrestrial water storage (TWS). Fields are plotted both as raw global average timeseries as well as "Delta" timeseries that show the running difference between consecutive 20-year averages. The latter show that TOTSOMC dominates TOTECOSYSC and is the slowest

term to equilibrate, but has reached a reasonable Delta value of around -0.04 Pg C after 4,000 years of spinup. The plots in the bottom row show that less than 10% of global land area remains outside of the target Delta range for TOTECOSYSC (shown as dashed lines in the Delta plots), and that the regions associated with slight disequilibrium at the end of spin-up are concentrated in northern Siberia.

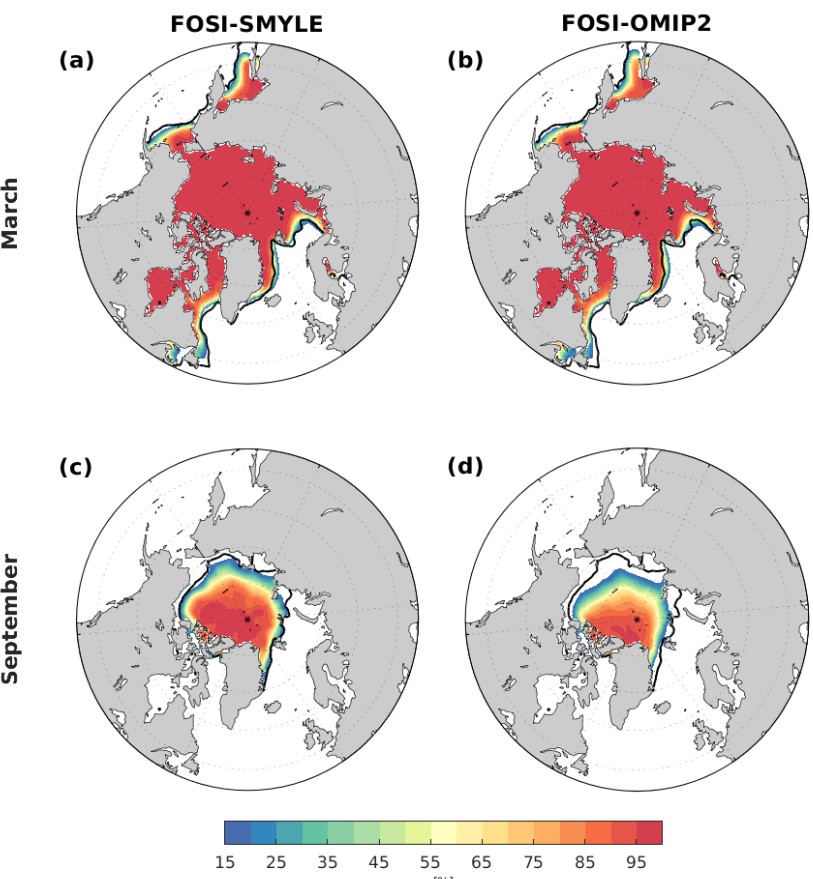

Fig. A2: Climatological (1991-2010) sea ice concentration for March (top) and September (bottom) from (a,c) SMYLE-FOSI and (b,d) FOSI-OMIP2. Black contour line shows the observed sea ice extent (15% concentration) from SSMI averaged over the same climatology. Note that the color scale starts from 15%.

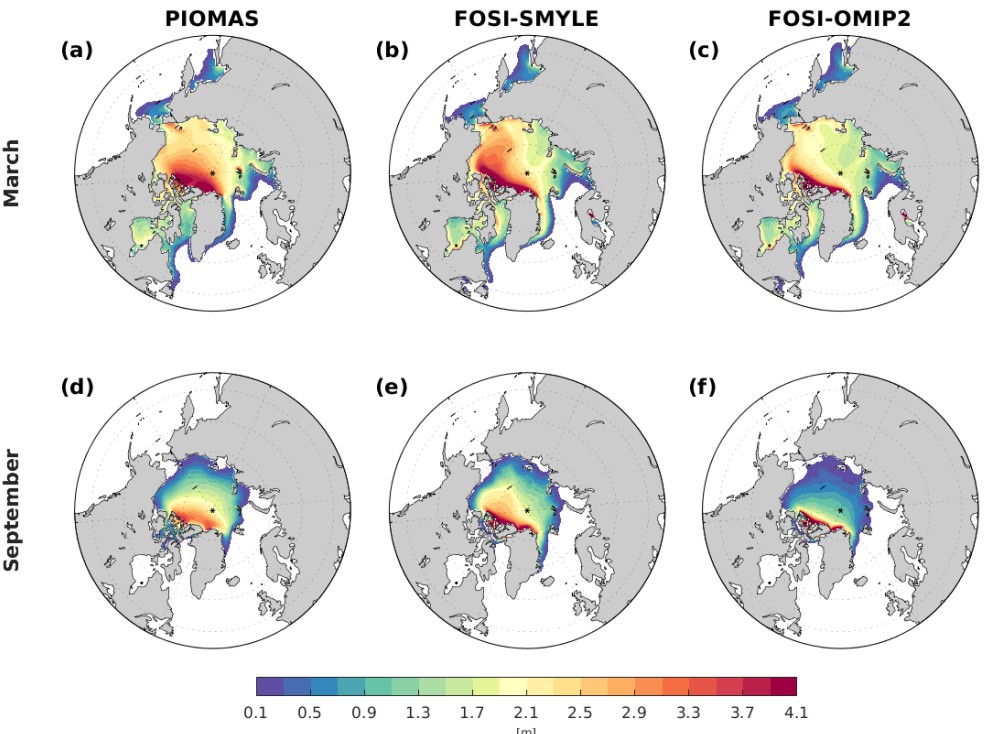

Fig. A3: Climatological (1991-2010) sea ice thickness for March (top) and September (bottom) from (a,d) PIOMAS, (b,e) FOSI-SMYLE, and (c,f) FOSI-OMIP2.

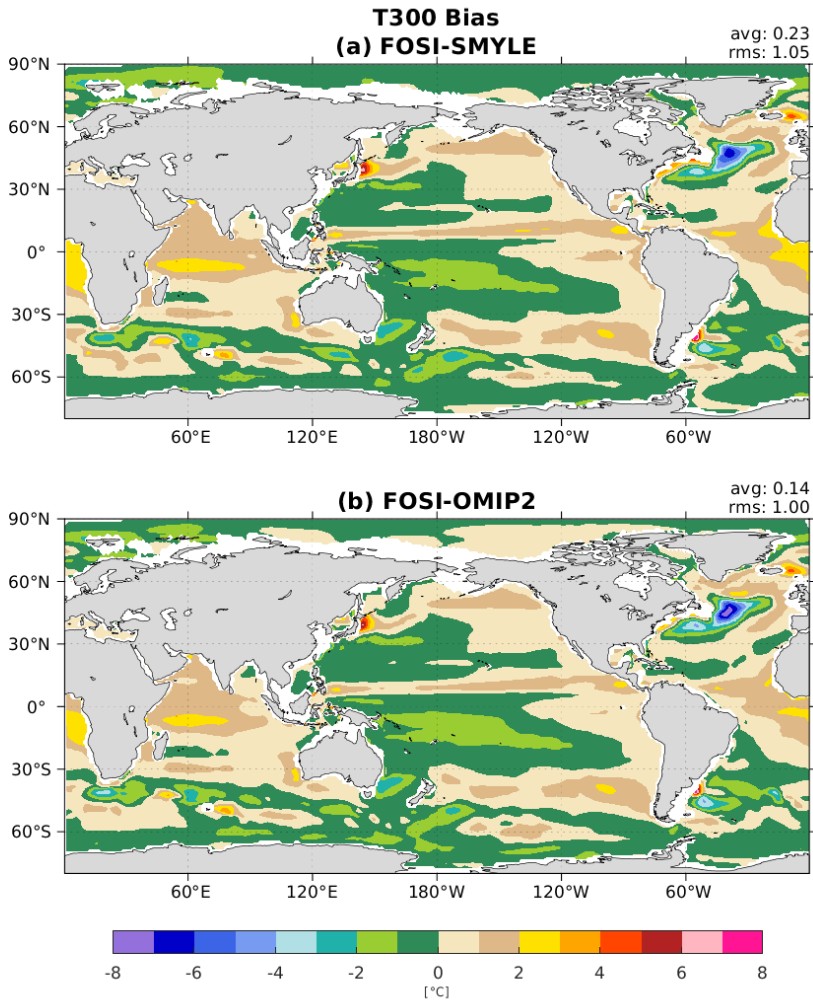

Fig. A4: Climatological (1991-2010) upper 300m ocean temperature bias relative to EN4 from (a) FOSI-SMYLE and (b) FOSI-OMIP2.


**Appendix B**

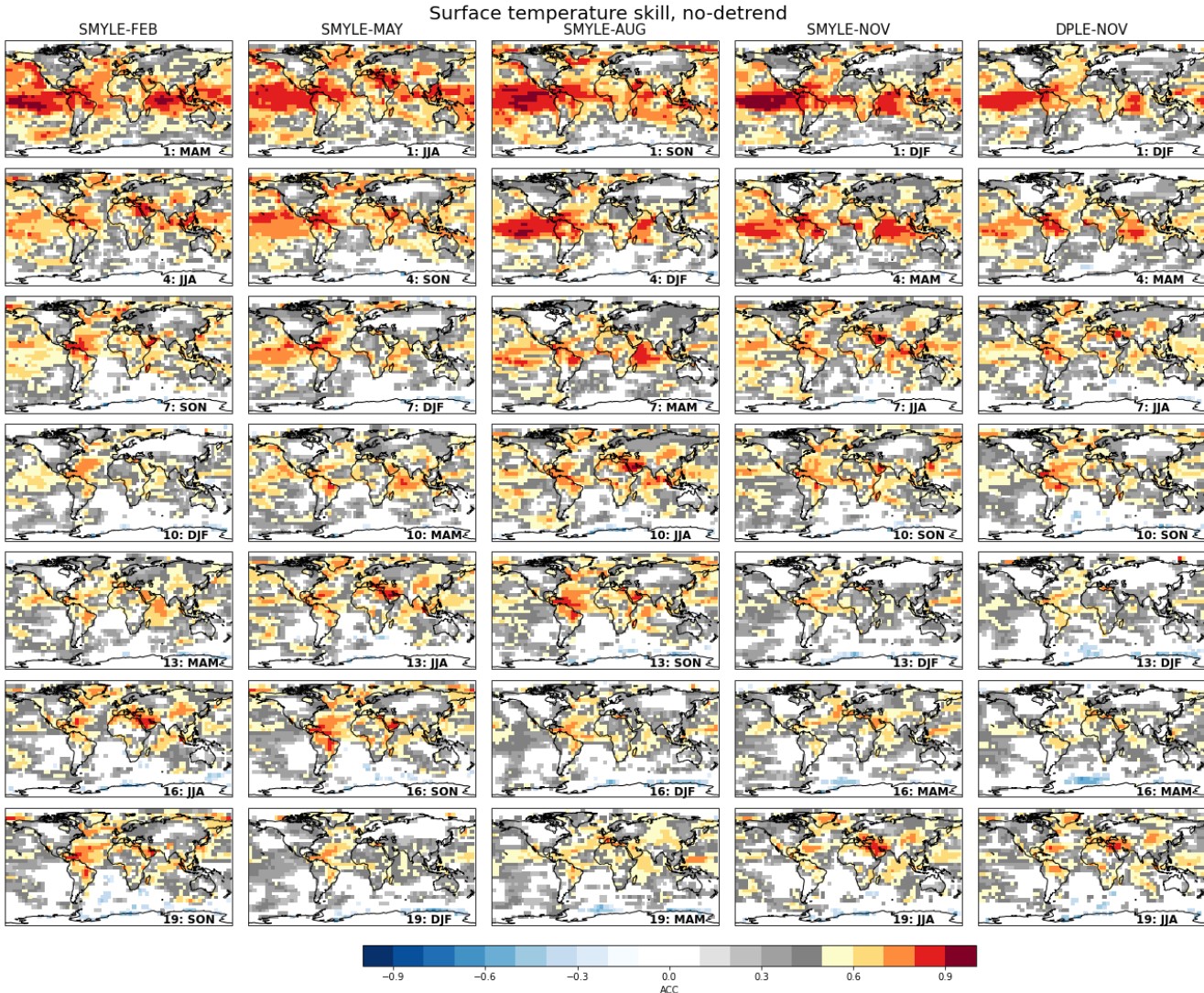

Fig. B1: Equivalent to Figure 1 (surface temperature ACC) but without removing a linear trend.

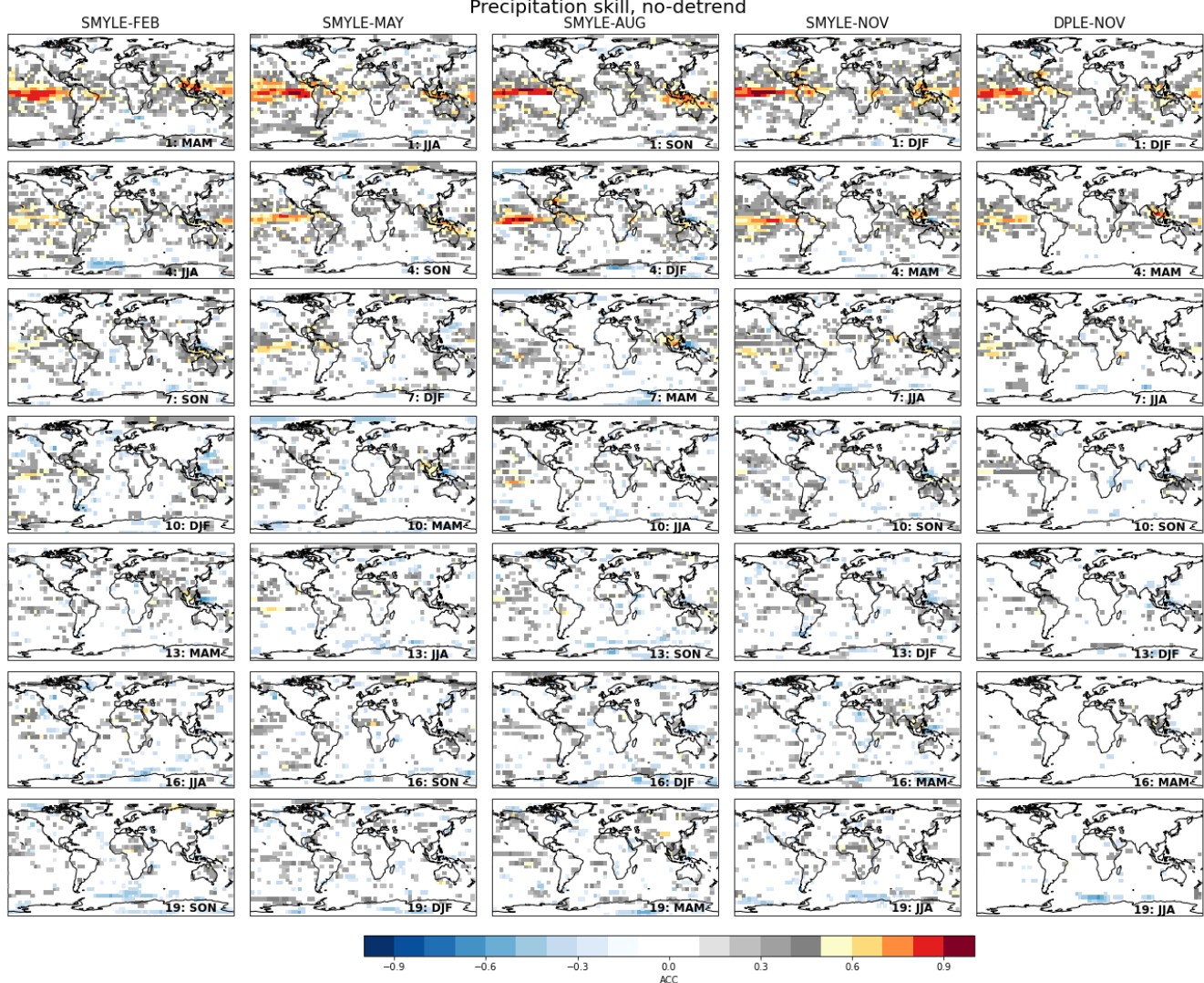


Fig. B2: Equivalent to Figure 2 (precipitation ACC) but without removing a linear trend.


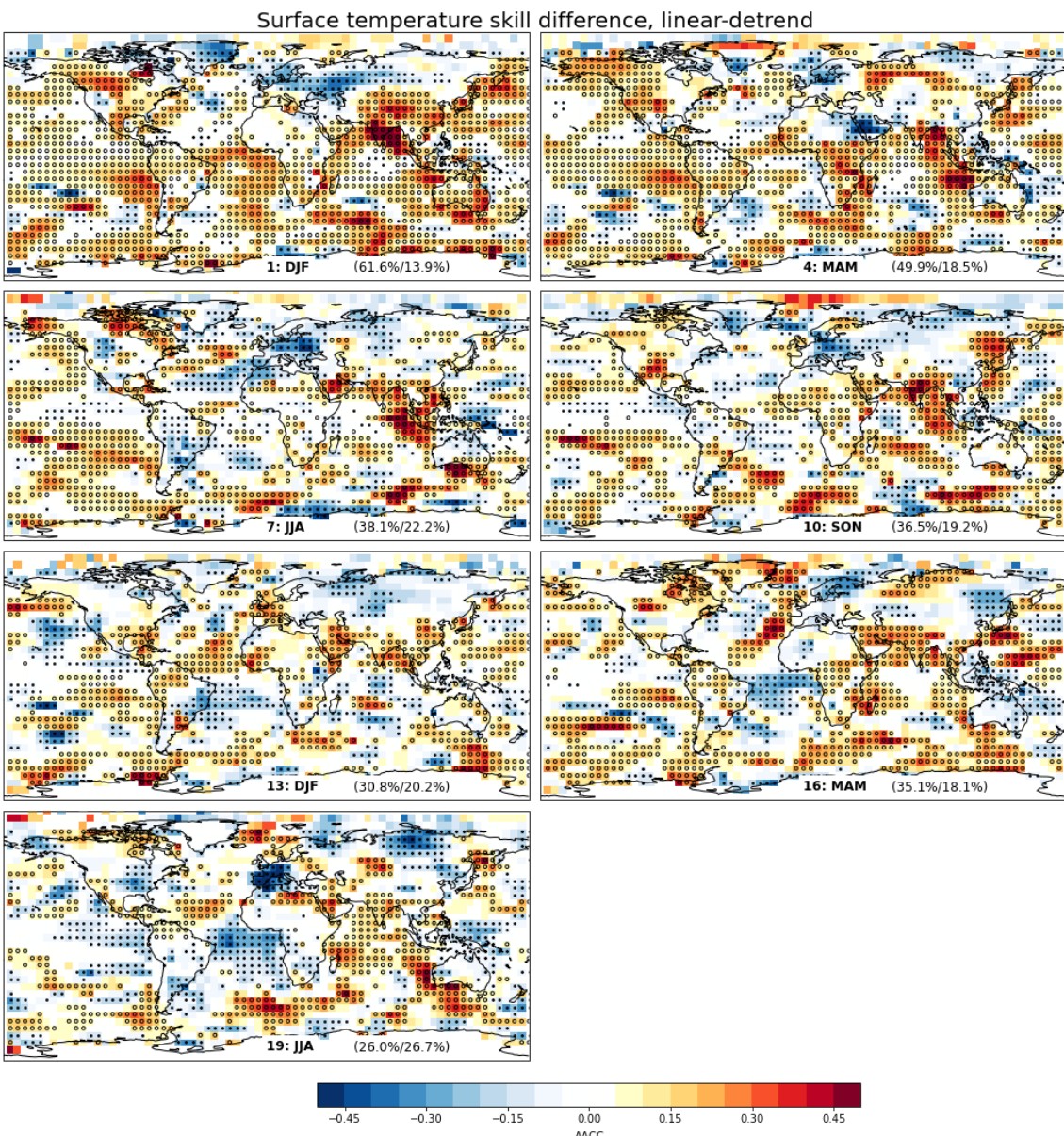

Fig. B3: Difference in ACC skill for surface temperature between SMYLE-NOV and the mean of the 20-member skill score distribution from DPLE-NOV. Here, both hindcast sets use identical verification windows (corresponding to start dates spanning 1970-2017). Open and filled circles indicate significantly higher and lower ACC in SMYLE-NOV, respectively (corresponding to SMYLE-NOV skill falling above the 90[th] or below the 10[th] percentile of a 100-member distribution of 20-member DPLE-NOV scores). Values in parentheses give the percentage of global surface area (within 80°S-80°N) where there is significant skill increase/decrease.

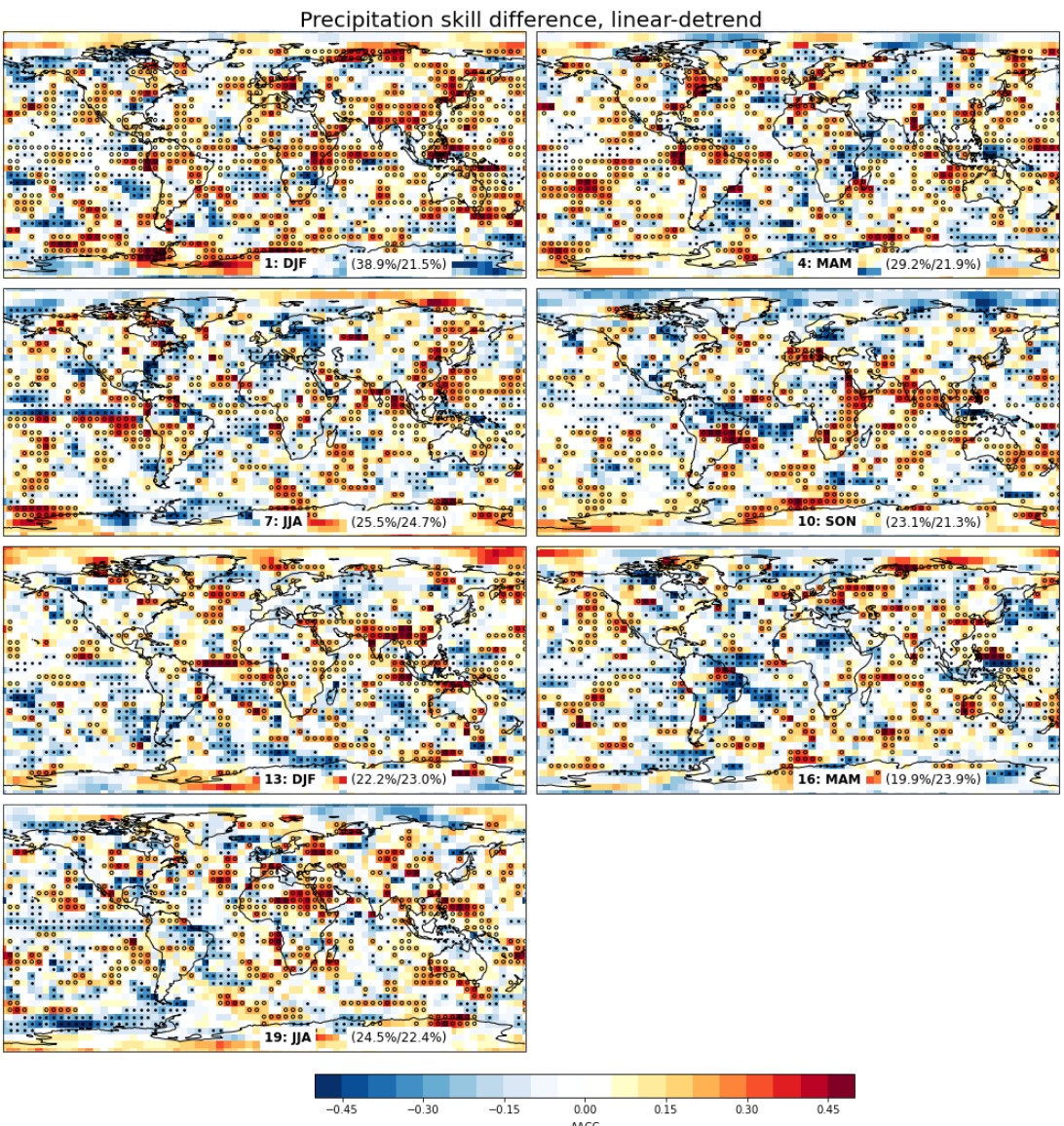

Fig. B4: Difference in ACC skill for precipitation between SMYLE-NOV and the mean of the 20-member skill score distribution from DPLE-NOV. Here, both hindcast sets use identical verification windows (corresponding to start dates spanning 1970-2017). Open and filled circles indicate significantly higher and lower ACC in SMYLE-NOV, respectively (corresponding to SMYLE-NOV skill falling above the 90th or below the 10th percentile of a 100-member distribution of 20-member DPLE-NOV scores). Values in parentheses give the percentage of global surface area (within 80°S-80°N) where there is significant skill increase/decrease.

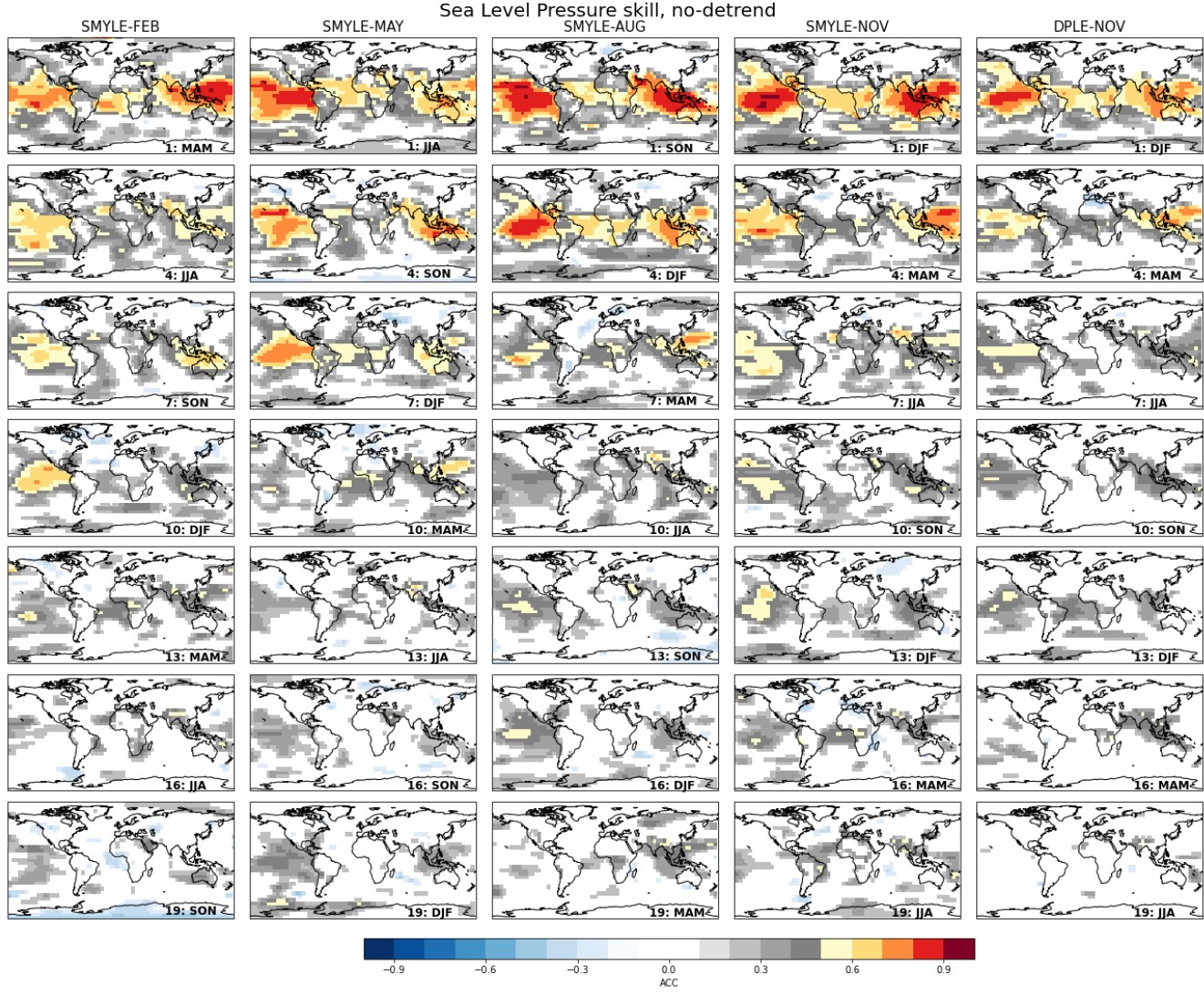

Fig. B5: Equivalent to Figure 6 (sea level pressure ACC) but without removing a linear trend.

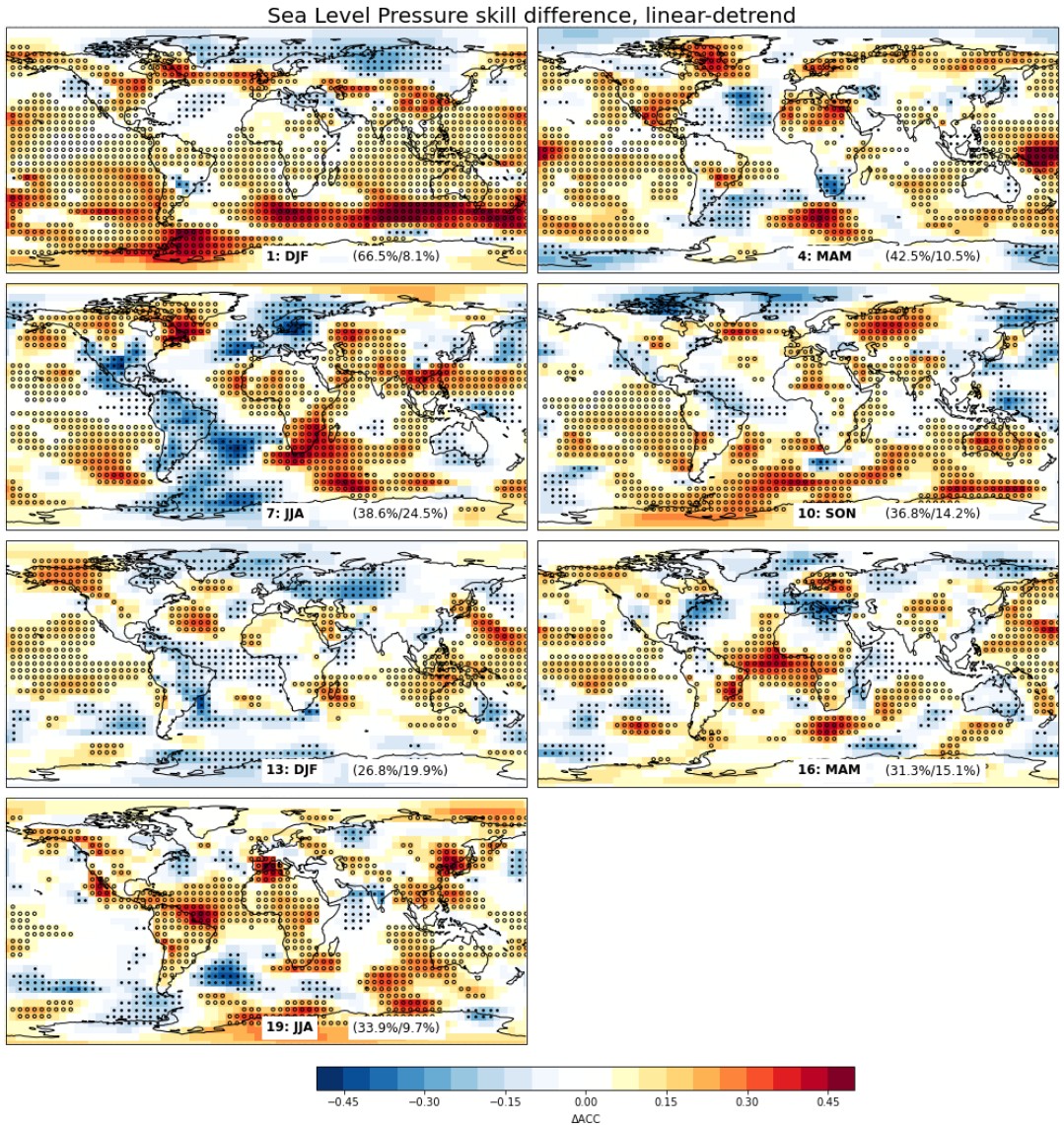

Fig. B6: Difference in ACC skill for sea level pressure between SMYLE-NOV and the mean of the 20-member skill score distribution from DPLE-NOV. Here, both hindcast sets use identical verification windows (corresponding to start dates spanning 1970-2017). Open and filled circles indicate significantly higher and lower ACC in SMYLE-NOV, respectively (corresponding to SMYLE-NOV skill falling above the 90[th] or below the 10[th] percentile of a 100-member distribution of 20-member DPLE-NOV scores). Values in parentheses give the percentage of global surface area (within 80°S-80°N) where there is significant skill increase/decrease.

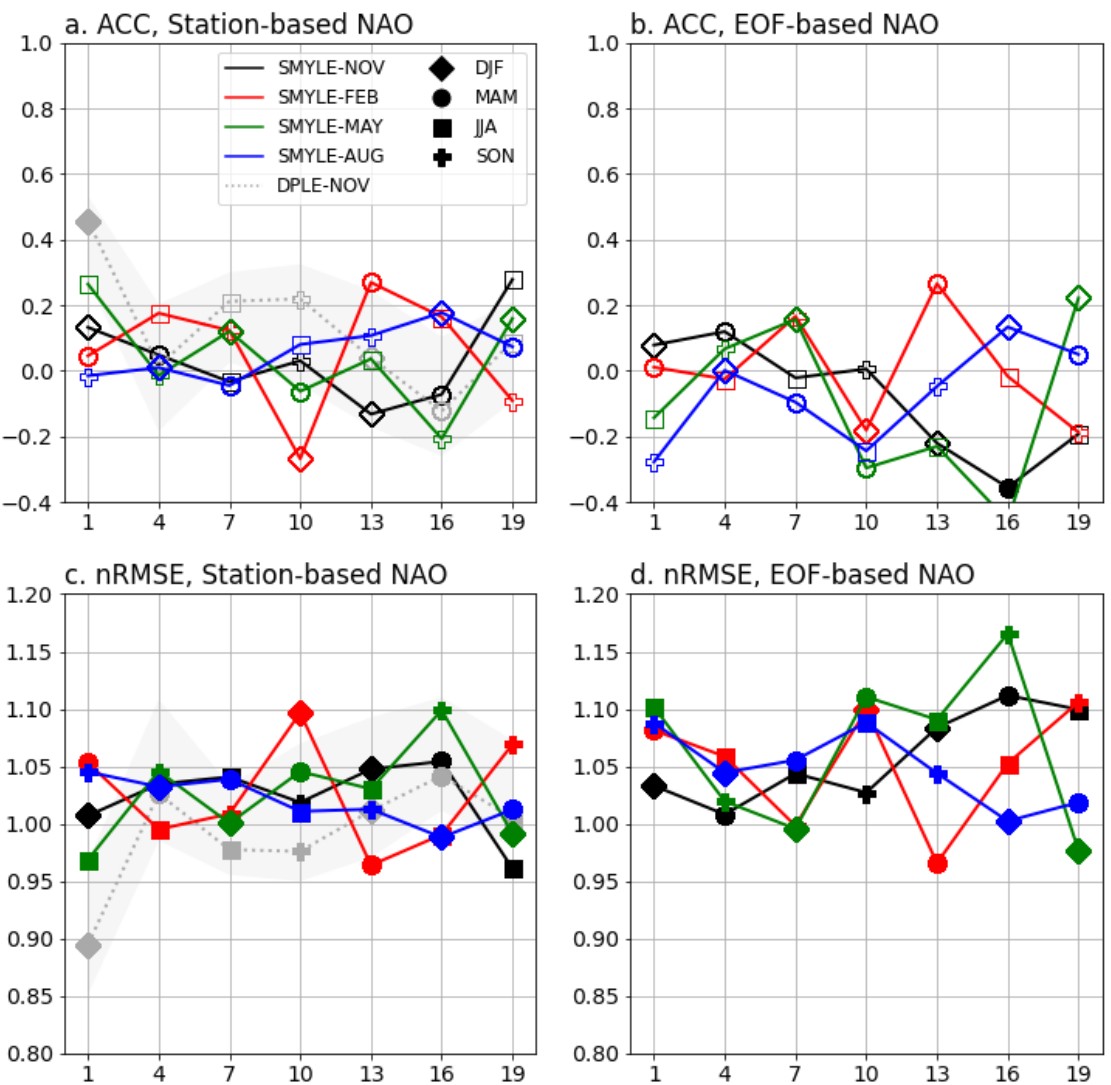

Fig. B7: Identical to Figure 7 except SMYLE and DPLE data are subsampled to only include forecasts initialized in years 1981-2015.

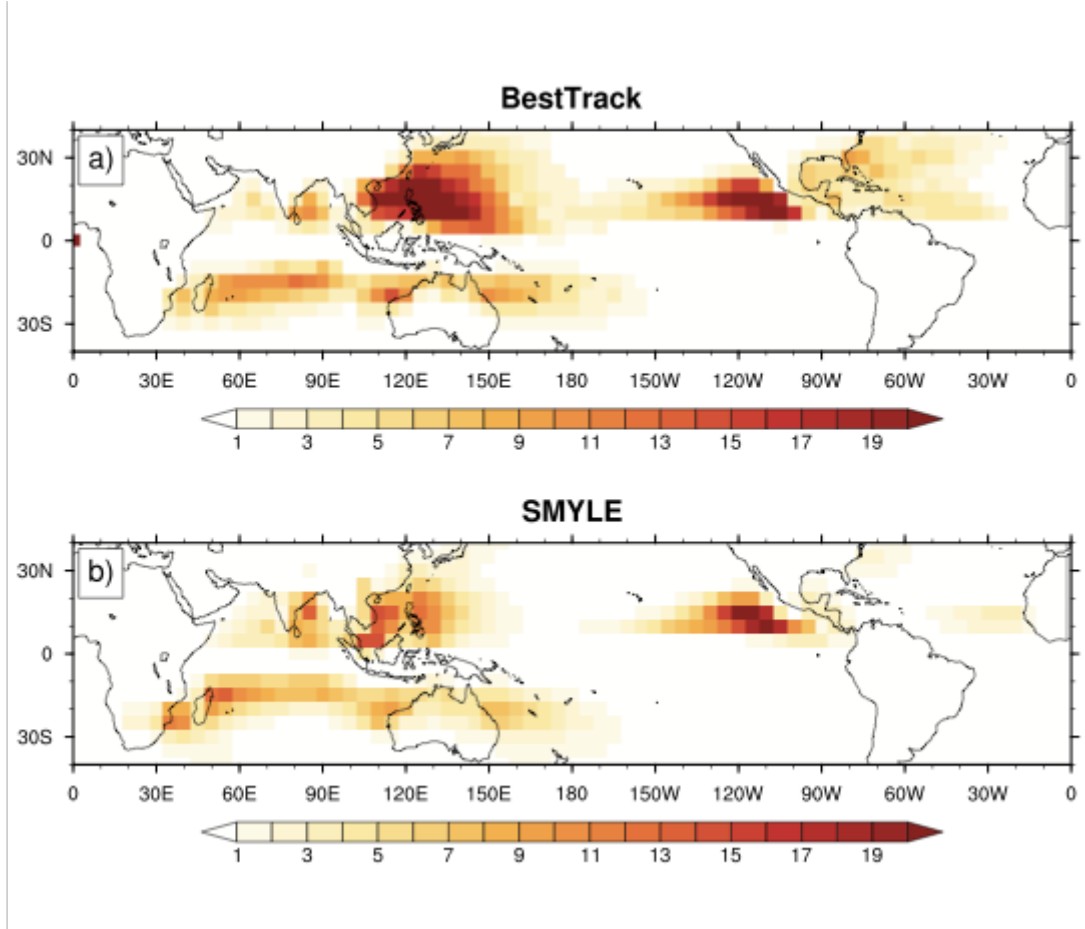

Fig. B8: Annual mean global TC track density (over 5˚x5˚ box) during 1970-2018 in (a) observations and (b) SMYLE lead month 1 forecasts (averaged over 4 initialization times).


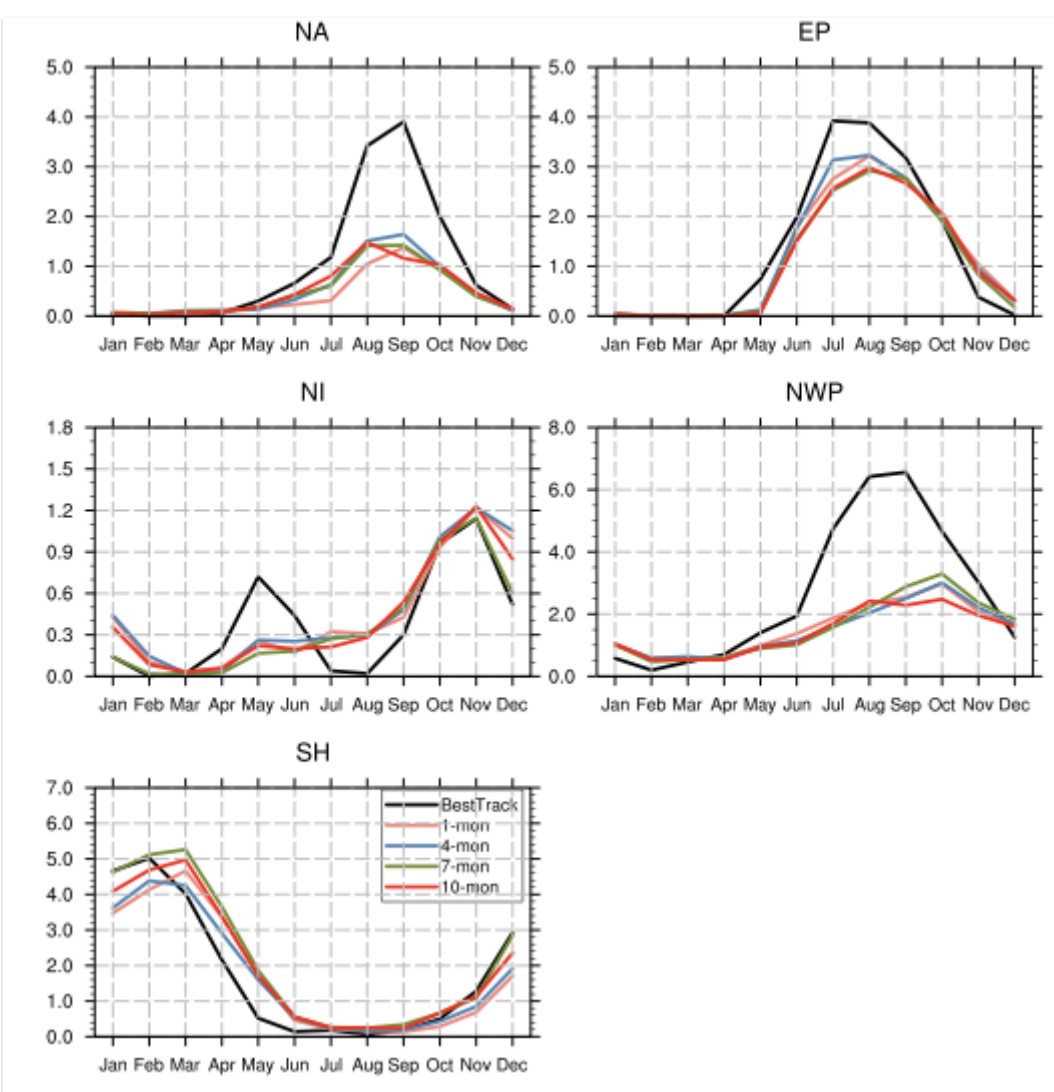

Fig. B9: Climatological mean TC number seasonality in observations (black curves) and SMYLE forecast at 1-month (orange), 4-month (blue), 7-month(green) and 10-month (red) lead time in different TC basins -- North Atlantic (NA), eastern Pacific (EP), North Indian Ocean (NI), Northwestern Pacific (NWP) and Southern Hemisphere (SH).



| Lead | NA | EP | NWP |
|---|---|---|---|
| 1-month | **0.61** | **0.3** | **0.25** |
| 4-month | **0.44** | 0.002 | 0.08 |
| 7-month | 0.14 | 0.06 | **0.28** |
| 10-month | 0.14 | -0.17 | 0.07 |
| 13-month | 0.22 | -0.16 | 0.07 |
| 16-month | 0.19 | 0.03 | 0.05 |
| 19-month | 0.2 | -0.02 | 0.09 |

Table B1: Similar to Table 3 of Main Text, but showing correlation coefficients of normalized seasonal-mean (JJASON) TC number between the observations and the SMYLE forecasts. Bold numbers indicate the correlations are significant at 90% confidence level. The correlations in the North Indian Ocean, the South Indian Ocean and the South Pacific Ocean are not significant at any leadtime and are therefore not shown.




*Code and data availability.* The SMYLE project webpage at https://www.cesm.ucar.edu/working-groups/earth-system-prediction/simulations/smyle includes pointers to SMYLE data and references, contact information, and instructions for how to replicate the experiment. Output data from SMYLE hindcast simulations as well as from the historical reconstructions used for initialization (the SMYLE FOSI and forced CLM5 runs) are available from NCAR's Climate Data Gateway at

https://doi.org/10.26024/pwma-re41. Output data from CESM2-LE simulations can be accessed at https://doi.org/10.26024/kgmp-c556. The model code, configuration files, and boundary condition data used for SMYLE (CESM 2.1) are all available from the CESM2 website: https://doi.org/10.5065/D67H1H0V. The analysis code (python, NCL, bash) along with auxiliary data used to generate the figures in this manuscript are available at https://doi.org/10.5281/zenodo.6341790. CRU-TS4.05 data were accessed from

https://crudata.uea.ac.uk/cru/data/hrg/cru_ts_4.05/. HadISST1 data were accessed from https://www.metoffice.gov.uk/hadobs/hadisst/data/download.html. JRA55 reanalysis data were accessed from https://esgf-node.llnl.gov/projects/create-ip/. ERA5 reanalysis data were accessed from https://www.ecmwf.int/en/forecasts/datasets/reanalysis-datasets/era5. GPCPv2.3 data were provided by the NOAA PSL, Boulder, Colorado, USA, from their website at https://psl.noaa.gov. OceanSODA-ETHZ data were accessed from

https://doi.org/10.25921/m5wx-ja34. NSIDC Sea Ice Index data were accessed from https://nsidc.org/data/seaice_index/. PIOMAS data were accessed from http://psc.apl.uw.edu/research/projects/arctic-sea-ice-volume-anomaly/data/. BestTrack data were accessed from https://doi.org/10.25921/82ty-9e16. NMME data were accessed from the IRI data library at https://iridl.ldeo.columbia.edu/SOURCES/.Models/.NMME/#info.


Author contributions. SGY designed the experiment and secured the required computing resources with assistance from JHR and GD. NR and AAG carried out the simulations. KL, DL, WW, WMK, GD, ML, DB, MH, and SGY all contributed to the generation of datasets used for initialization. SGY, XW, IS, HL, MJM, KK, SM, NL, and WW contributed to the analysis of SMYLE output. SGY prepared the manuscript with contributions from all authors. SGY created the repository of

python analysis codes with help from IS, MJM, KK, XW, WW, and TK. WGS assisted with data management.

*Competing interests.* The authors have no competing interests.

*Acknowledgments.* The CESM project is supported primarily by the National Science Foundation (NSF). The National

Center for Atmospheric Research (NCAR) is a major facility sponsored by the NSF under Cooperative Agreement 1852977.

Portions of this study were supported by the Regional and Global Model Analysis (RGMA) component of the Earth and Environmental System Modeling Program of the U.S. Department of Energy's Office of Biological and Environmental Research (BER) via NSF Interagency Agreement 1844590, and by Department of Commerce Grant NA20OAR4310408. Computing and data storage resources, including use of the Cheyenne supercomputer (doi:10.5065/D6RX99HX), were provided by the Computational and Information Systems Laboratory (CISL) at NCAR. More than half of SMYLE simulations were made possible by an NCAR Strategic Capability (NSC) award from CISL; the remainder utilized the Climate Simulation Laboratory (CSL) allocation of the CESM Earth System Prediction Working Group. We thank all the scientists, software engineers, and administrators who contributed to the development of CESM2.

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
