# Peer review of "The Seasonal-to-Multiyear Large Ensemble (SMYLE) Prediction System using the Community Earth System Model Version 2"

_Geoscientific Model Development, 2022_

## Author Response (AR1)

**Author Response**

Referee comments are repeated below in italics followed by our point-by-point responses. General modifications to note include: 1) elimination of supplemental material (extra figures are now included in appendices) and 2) all main figures are now embedded in the text near where they are first discussed. All line number references below refer to the marked-up version of the revised manuscript.

**RC1**

This manuscript describes and evaluates a new ensemble prediction system for lead times up to 2 years. The system is based on the previously documented CESM2 (Danabasoglu et al. 2020) Earth system model, initialized from JRA-55 in the atmosphere, a forced integration following the OMIP2 protocol for ocean and sea ice (FOCI), and a forced simulation of the CLM land surface component. 20-member ensembles are initialized for four initial months per year over the 1970-2019 period, making this dataset a substantial contribution to the climate prediction community.

The paper is well organized, pleasant to read and instructive. I acknowledge the thorough effort led by the authors to evaluate a more diverse range of variables and indices beyond sea surface temperature, circulation indices and precipitation, thereby illustrating the interest of such a system based on an Earth system model and promoting further research with this database. The goals of the paper are clearly stated at the end of the introduction, and in my view, rather adequately fulfilled in the following sections. The number of figures remains quite reasonable with respect to the completeness of the analysis.

Given the quality of this submission, I recommend to accept it for publication in GMD, subject to minor revisions. I have two main points I wish to raise, and more specific comments and minor suggestions follow.

> Thank you for the overall positive assessment and many helpful suggestions.

1) Although the initialization strategy is described in detail, the authors focus the evaluation on the seasonal-to-multiyear forecast skill. For some variables for which observational data is scarce, or does not cover the entire hindcast period, reconstructions used for initialization are also used as a reference for skill assessments. I understand the reason for this choice but would then have expected more details on the estimated quality of these reconstructions when these haven't been documented elsewhere (which is the case at least for FOSI). For instance, the authors mention some shortcomings of the CESM2 contribution to OMIP2 that were corrected by tuning parameters and restoring strength for FOSI, but no further details or evaluation of the improvement with respect to independent estimates are provided (it could be included as supplementary information).

> We have added four new figures in new Appendix A (Figs. A1-A4) that serve to document the reconstructions used for initializing ocean/sea-ice and land in SMYLE. Figures A2-A4 show how the FOSI used for SMYLE compares to that submitted to OMIP2 in terms of Arctic sea-ice and upper ocean temperature. We added several new lines to the discussion of SMYLE FOSI (Section 2, starting line 149). Likewise, the description of the land-only simulation has been

modified/expanded to include reference to a new figure showing the equilibration of land carbon pools (Fig. A1) as well as additional literature references that serve to document simulation quality (Section 2, starting line 134).

With respect to these reconstructions, were any long-term drifts found? I acknowledge several cycles of the forced model have been run but is it enough to avoid spurious effects in the hindcasts?

Standard diagnostics performed on SMYLE FOSI revealed low levels of drift in this simulation for (near) surface fields after cycle 3, on par with those documented in Tsujino et al. (2020) for the CESM-POP OMIP2 experiment. As noted above (and now in the text, line 152), the change in diffusion parameter significantly reduced the ocean drift below 2000m, which is generally where the largest magnitude long-term drifts are found in OMIP simulations (Tsujino et al. 2020).

2) Furthermore, although some figures provide an indication of the ensemble spread, skill is evaluated solely using deterministic scores (anomaly correlation coefficients of the ensemble mean, root mean square error). Having a 20-member ensemble allows for the assessment of other aspects of forecast quality, including reliability and resolution, or other probabilistic metrics of hindcast skill.

We completely agree with the reviewer that it would be very desirable to see an assessment of SMYLE skill using probabilistic metrics. We feel, however, that such an evaluation would be best done in a separate, dedicated study that could complement the deterministic skill assessment presented here. The manuscript is already long, and inclusion of even one example probabilistic analysis would require rather lengthy discussion and explanation of methods. We've added a sentence in the conclusion noting the need for a full probabilistic assessment in the future.

Specific comments

1) The authors compare some skill assessments with other reference systems such as the NMME multimodel (for seasonal time scales) and CESM1 DPLE (for November initializations). However these comparisons are only shown in a selection of figures. I'm not necessarily asking for comparisons to be included in each figure, but more discussion on similarities / discrepancies in skill with these two benchmarks could be of interest to the reader.

We have added an Appendix B that includes and expands upon the supporting figure set previously included as supplemental material. Several new figures included in Appendix B provide more detailed information on the SMYLE-NOV vs. DPLE-NOV skill comparison. Specifically, the descriptions of seasonal skill for surface temperature and precipitation (Section 3.1) and seas level pressure (Section 3.3) have been augmented with discussion of Appendix B plots that show a more rigorous SMYLE/DPLE comparison (skill difference plots with bootstrapped significance testing). We also now include DPLE in the NAO analysis (Section 3.3; see below for details) and associated discussion. Finally, we've added discussion of the Befort et al. (2022) results for seasonal TC forecasts in Section 3.7 (see response to comment below).

2) I was confused by differences in lead time values in Figure 5 and in the text (lines 298-312). The shorter lead months don't seem to appear on the plots although they are mentioned in the text (ie: line

300 refers to an ACC of 0.65 I cannot find on the plot). Furthermore using the color and symbol code, lead month 2 for SMYLE-FEB reads as JJA, which isn't consistent at all with definitions provided in the paragraph starting at line 164 – and doesn't make sense. Could you please revise the figure?

>We've revised Figure 5 and corrected some errors in the discussion of this figure.

3) Correlation / ACC values are often referred to as significant / non-significant, but I found no mention of the significance test and underlying hypotheses (sorry if I missed it!).

>In response to this comment and a request from RC2 for a definition of nRMSE, we have added a paragraph to Section 2 that clarifies the skill metrics used in the paper along with the methods for evaluating significance (Section 2, starting line 202).

Minor suggestions

l. 46: "seasonal protocols call for ensemble simulations lasting 12 months" → not all operational systems go up to 12 months; by WMO standards, seasonal prediction information is provided up to ~ 6 months. I would recommend saying "lasting up to 12 months".

>Done.

l. 65-69: Some of the potential sources of predictability are associated to a reference, whereas others are not; I would recommend harmonizing this. For snow cover: consider Orsolini et al. (2016) or more recently Ruggieri et al. (2022). For QBO: consider Butler et al. (2016) (QJRMS). For greenhouse gas forcing: Doblas-Reyes et al. (2006)

>Done. Thanks for these suggestions.

l. 181: JRA55 (reanalysis) data for precipitation is not an obvious choice; is this due to the hindcast period? Couldn't you use merged precipitation datasets such as GPCP which probably have a higher fidelity to actual observations?

>Thanks for this suggestion to improve the precipitation skill assessment. We now use GPCP v2.3 (1979-2021) as the precipitation verification dataset. This resulted in some slight changes in the skill maps, and so the discussion has been modified accordingly (Section 3, starting line 250).

l. 310: Not unrelated to my earlier comment on assessing probabilistic skill and using the 20-member ensemble, did you evaluate the ensemble spread of SMYLE according to target season and forecast time for these ocean indices?

>See response to main comment above. These ocean indices will be good targets to include in a future probabilistic skill study.

l. 348-364: Was this low (no) NAO skill already found with DPLE-NOV? Another aspect, beyond horizontal and vertical resolution of the atmosphere, is the sensitivity of correlation of NAO to the ensemble size, the length of the re-forecast period (see e.g. Shi et al., 2015) and low-frequency

variability of NAO skill during the last century (Weisheimer et al., 2019). They suggest that RMS-based scores are less sensitive estimates of NAO skill.

> We have substantially revised the Section 3.3 discussion of NAO skill. Revised Figure 7 now includes DPLE-NOV results (both 40-member mean skill as well as 20-member skill spread), and we've added panels showing nRMSE scores. The new plot shows that SMYLE NAO skill is lower than DPLE NAO skill, but not significantly so across all lead times. We've also added Figure B7 that replicates Figure 7 but using a shorter forecast time window (1982-2015) to demonstrate sensitivity to verification window. We now cite both Shi et al. (2015) and Weisheimer et al. (2019) in this section (see paragraph beginning at line 518).

l. 389: I'm not at all a biogeochemistry expert; there appears to be some variability in skill according to the target season, with summer and fall Zoo C, NPP and carbon export more predictable than winter or spring. Why is this the case? What are the drivers behind what appears to be a return of potential predictability in SMYLE? Some discussion (or references) on this would be helpful!

> The return of potential predictability in these ocean ecosystem variables during summer/fall is indeed an interesting phenomenon and we thank the reviewer for pointing this out. We think that this is due to the wintertime reemergence of subsurface nutrient anomalies that were present during initialization. These nutrient anomalies then drive anomalies in ecosystem productivity the following summer and fall. This mechanism is well-described in Park et al. (2019, Science), who observed the reemergence of chlorophyll prediction skill during the summer and lower prediction skill during winter. In response to this comment, we have added a few additional sentences at the end of this paragraph to explain this mechanism (starting at line 588).

Figure 9 is a bit blurry: could you increase its resolution?

> Done.

In figures 10 and 11, correlation and RMSE for CESM2-LE are plotted at lead month 19. I find this choice confusing since CESM2-LE is not initialized; maybe you could use a dotted or dashed line as done for the persistence forecasts?

> Done.

l. 465-480: Summer (JAS) SIE trends in SMYLE seem different from FOSI, with the ensemble mean generally below FOSI values in the 1970s-1980s, and above after the mid-2000s. Do you have an explanation for what appears to be conditional drift? Did you compare the sea ice thickness fields in SMYLE with FOSI?

> We do not have an explanation for conditional bias in summer SIE in SMYLE, and it would likely require a more in-depth examination of the spatial distribution of anomalies (as a function of initialization year) to offer more insight. This is beyond the scope of the present study. We have not compared sea ice thickness, but we do compare sea ice volume (Fig. 13) which does not show the conditional bias. We've added the following sentence to the sea ice discussion (Section 3.6; starting at line 751):

"It is interesting to note that the conditional bias seen in SMYLE for JAS SIE (resulting in a lower decreasing trend than seen in observations or FOSI, particularly at long leads; Fig. 12), is not evident for JAS SIV even at lead month 20 (Fig. 13). The explanation for this merits further investigation, but it implies there is a compensating conditional bias in the sea ice thickness field."

Section 3.7: Results are interesting, however some comparison with other recent evaluations would be nice. Although not focusing on the same period, and using IBTrACS as a reference, Befort et al. (2022) (their Fig. 5) would be a nice comparison for your lead time 1 month results in table 1.

Thank you for this suggestion. We've added the following in Section 3.7 (starting at line 841):

"Skill for the NA and NWP regions at 1-month lead is generally comparable with other seasonal forecast models. For example, Befort et al. (2022) evaluated TC prediction skill over the NA (during JASO) and NWP (during JJASO) for the period of 1993-2014 in six seasonal forecast systems. They found that the models on average have a correlation coefficient of 0.6 for ACE over the NA.  For the NWP, the average correlation is 0.65, with 0.4 being the lowest value. Despite having a lower model resolution, SMYLE skill falls within the range of these results, although the comparison is complicated by different verification windows and different definitions of active TC season."

Fig. 15: This figure is quite difficult to read and interpret as it superimposes many time series. I would suggest either presenting a subset of information, or including it in the supplement to the article.

We have simplified Figure 15 to highlight only lead month 1 as well as the longest lead time that yields a significant correlation with the observations (as shown in Table 1).

l. 560: missing word ("as")? "as well an experimental system"

Fixed.

l. 574: Out of curiosity: are there any plans to update the system in near real-time? How frequently is JRA55-do updated?

We do mention in the concluding paragraph that:
"The choice to use JRA55 (JRA55-do) as the basis for component state reconstruction means that SMYLE could potentially be extended back in time as far as 1958, and forward in time to near real-time. We anticipate that future SMYLE extensions will further enhance the utility of this resource."

JRA55-do is officially updated annually, but we have tools that permit near real-time updates of JRA55-do based on weekly updates of the base JRA55 reanalysis. We prefer to leave the manuscript text rather vague on this point, as NCAR is not committed to supporting the generation of real-time forecast products. SMYLE is primarily a research tool, not an operational forecasting tool.

References mentioned:

Befort et al. (2022) doi: 10.1175/JCLI-D-21-0041.1
Butler et al. (2016) QJRMS 142(696):1413–1427
Doblas-Reyes et al. (2016) doi: 10.1029/2005GL025061
Orsolini et al. (2016) Climate Dynamics 47(3–4):1325–1334
Ruggieri et al. (2022) doi: 10.1007/s00382-021-06016-z
Shi et al. (2015) doi: 10.1002/2014GL062829
Weisheimer et al. (2019) doi: 10.1002/qj.3446
Citation: https://doi.org/10.5194/gmd-2022-60-RC1

**RC2**

This submission describes an extensive set of hindcasts from the CESM2 model that enable the performance of initialized predictions in the relatively unexamined multi-year range (out to 24 months in this case) to be extensively explored. Notably, performance over a broad range of Earth system components (atmosphere, sea ice, ocean and land including biogeochemistry) is addressed. The paper is very well organized and written, and criticisms are limited mainly to relatively minor details of description and presentation. Exceptions are items 7 and 17 below, which will require modest additional computation if the authors concur that acting on these recommendations will improve the paper. Overall however the authors are to be congratulated for this interesting and compelling documentation of SMYLE.

> Thank you for the overall positive assessment and many helpful suggestions.

Main comments

1) At line 47, suggest replacing "at least 10-years duration" with "up to 10-years duration" because some operational "decadal" systems have a 5-year range, but none that I'm aware of run for >10 years.

> Done.

2) Suggest additionally referencing Boer et al. https://doi.org/10.1007/s00382-013-1705-0 and Chikamoto et al. https://doi.org/10.1038/s41598-017-06869-7 in the sentence starting on line 66, possibly as follows: "…volcanic activity (Hermanson et al. 2020), greenhouse gas forcing (Boer et al. 2013), or some combination thereof (Chikamoto et al. 2017)."

> Done. Thanks for these suggestions.

3) At line 72 should also reference Ilyina et al. https://doi.org/10.1029/2020GL090695

> Done.

4) Line 128 states that forcing is applied cyclically form 1901-1920 to equilibrate the land state. Please go into a bit more detail about the total length of time this cyclic forcing was applied, in relation to expected equilibration times of land variables such as vegetation and soil carbon.

> In response to this comment as well as a similar comment from RC1, we have added Figure A4 to the new Appendix A. This figure demonstrates the near equilibration of land carbon stocks over ~4,000 years of spin-up simulation. Despite some slight imbalance in total ecosystem carbon (primarily associated with soil organic matter carbon), the spin-up was deemed close enough to equilibrium to proceed with the land-only historical simulation. The associated discussion in the text has been expanded (Section 2, starting at line 134).

5) It's stated that the hindcasts cover 1970-2019. Presumably this is the period covered by the initialization times, and not the simulations themselves which would extend into 2021? If so please be explicit that 1970-2019 spans the initialization times.

> We've modified the text throughout to clarify that mention of specific historical windows (e.g., 1970-2019) refers to forecast initialization years included in the analysis. We also added a discussion in section 2 clarifying that the verification window is a function of: 1) hindcast initialization time, and 2) observational temporal coverage. Our general approach is to maximize temporal sampling to the extent possible, which means that the actual verification window can vary with lead time. We think this is clear while also avoiding burdening the reader with excessive detail.

6) Regarding "Potentially useful prediction skill (ACC>0.5) is seen for land precipitation over the southwestern US in DJF and MAM (lead month 1)" at lines 208-209, this really should say "southwestern North America" considering that the only DJF grid boxes >0.5 are in Mexico.

> In response to a comment from RC1, we have changed the precipitation verification dataset to GPCPv2.3 (spanning 1979-2021). This has resulted in slight changes in skill scores and regions/leads where ACC>0.5. The discussion of Figure 2 has been revised accordingly, and in particular, we now highlight "southwestern North America in DJF (lead month 1)".

7) Regarding "A more rigorous analysis is needed to definitively demonstrate that SMYLE skill differences from DPLE are statistically significant and not likely explainable by chance" (lines 226-227), this could be done relatively easily by applying the random walk methodology of DelSole and Tippett, https://doi.org/10.1175/MWR-D-15-0218.1, where differences either in the anomaly pattern correlation or the RMSE between the 50 pairs of November-initialized hindcasts could be used as the basis for comparison.

> Thanks for this suggestion. In response to RC1 and this comment, we chose to include additional figures in Appendix B (see Figs. B3, B4, B6) that test the significance of local ACC differences between SMYLE-NOV and DPLE-NOV by accounting for uncertainty due to finite ensemble size. The 20-member SMYLE-NOV skill is compared to a distribution of 20-member DPLE-NOV skill scores at each grid point, and overall performance is assessed by comparing the percentage of global surface area (within 80°S-80°N) associated with skill improvement/degradation. Other methods (such as that proposed by DelSole and Tippett) could

be explored in future work, but we think simple skill difference plots will be of particular interest to readers and will suffice for this manuscript.

8) At line 178, please provide a rationale for regridding to a 5x5 or 3x3 degree grid. (Also a small point, but I'm not sure that regridding to a coarser grid qualifies as "interpolation".)

> We have replaced "interpolated" with "mapped" and have added the following rationale (line 198):
> "This remapping is done to highlight aggregate regional skill, increase the efficiency of skill score computation, and improve the quality of global map plots that include pointwise significance markers (Appendix B)."

9) Below line 360 it would be appropriate to reference Butler et al. https://doi.org/10.1002/qj.2743 in relation to the influence of lid height on skill in forecasting the NAO. (For example could append as "…relative to these baseline SMYLE results, although a robust connection between atmospheric vertical resolution and NAO skill has not been demonstrated (Butler et al. 2015).")

> Done.

10) Please replot Figs. 8d-i using the tick mark values in Fig. 9 which are better aligned with the experiment.

> Done.

11) Are there any evident explanations or hypotheses for the strong seasonal dependence of skill in Figs. 8d-i, e.g. high SE US shelf NPP ACC in JJA and SON, and low CA current NPP ACC in DJF?

> We think that the seasonal dependence of potential predictability skill in these regions concerns the persistence of subsurface nutrient anomalies (first described in Park et al., 2019). During wintertime mixing, these nutrient anomalies reemerge in the upper ocean and contribute to skill in predicting total summer/fall productivity in these LMEs. We agree that this is interesting and therefore, in response to this comment, we have added a few additional sentences at the end of this paragraph to elaborate on the seasonal dependence of marine ecosystem predictability (starting at line 588).

12) Should mention in the captions for Figs. 8-9 that shading and filled symbols indicate statistical significance.

> Done.

13) The OceanSODA-ETHZ aragonite saturation dataset covers 1985 to 2018 according to Gregor and Gruber (2021), so presumably the skill results in Fig. 9 are specific to this period? Or does the verification period leave out the years before 1990 which are much more uncertain according to those authors? Please be explicit about this and any other deviations of verification periods from the 1970-2019 period covered by SMYLE.

The verification against OceanSODA-ETHZ is performed over the window 1985-2018. We have clarified this in the text (line 608). We've also made efforts to be clear throughout about verification windows, without getting into excessive detail (see response to comment above).

14) In the captions to Figs. 10 and 11 suggest removing "(see text for details)" since the text doesn't provide any significant additional detail.

Done.

15) In Fig. 13 the lead times in the legends of the plots disagree with the lead times indicated in the caption.
16) Relating to Figs. 12 and 13, it would be interesting to have a sense of how the correlation and nRMSE values shown compare to values based on comparing OBS to FOSI.

We have corrected Figure 13 to accurately reflect the lead times mentioned in the caption (2, 8, 14, 20), and we've removed the sea ice skill scores from plots (Figs. 12, 13) and put them in new tables (Tables 1,2). We think the new tables help improve readability, and they facilitate inclusion of scores between OBS and FOSI.

17) Figure 14 shows JJASON (NH) and DJFMMA (SH) cyclone track densities regressed against annual mean Nino3.4 index. However, because ENSO typically peaks around December and frequently changes phase between about April and August, annual mean Nino3.4 is not a very good indicator of ENSO activity. In addition, this procedure introduces a seasonal disconnect in that January Nino34 is presumed to influence TC activity in the following November (for example). Suggest instead regressing JJASON track densities against JJASON Nino3.4, and DJFMAM track densities against DJFMAM Nino3.4, or else better justifying the original choice made. (Also please be explicit in the caption to Fig. 14 what is the timing of the Nino3.4 index.)

We have updated Figure 14 to show TC track density regressed against the respective seasonal mean Nino 3.4 index (i.e, JJASON track density regressed against JJASON Nino 3.4 index for the Northern Hemisphere, DJFMAM for the Southern Hemisphere). Compared to the annual mean Nino 3.4 index, the seasonal Nino 3.4 index yields a slight decrease of the positive regression coefficients at 10-month lead and longer. The main conclusions remain unchanged.

18) Tables 1 and S1 along with Fig. 15 imply that JJASON and DJFMAM TC predictions are made at 19-month lead time. However, for JJASON this implies initialization on 1 Nov, which in turn implies prediction of JJASO (not JJASON) at 19-month lead by the 24-month hindcast, and similarly for DJFMAM. Although this is a small point it should briefly be acknowledged somewhere (similarly to the 22-mon lead Nino3.4 forecasts in the caption to Fig. 4).

We have added this clarification in the captions of Figs. 14, 15 as well as the TC ACE correlation table (now Table 3).

19) Regarding the RMSE scores, "RMSE" isn't defined anywhere, and when introducing RMSE in the text should briefly comment on the use of normalized RMSE and introduce the nRMSE notation. Also, is nRMSE defined such that predictions of climatology (zero anomaly) will yield values of 1? If so then

briefly mentioning this will help the reader appreciate that nRMSE values <1 indicate that the prediction is more skillful than a climatological prediction.

A new paragraph in section 2 (starting line 202) introduces the skill metrics examined in the paper. We note that nRMSE is defined such that a climatology forecast yields a value of 1.

Technical corrections:

line 203: central America -> Central America

Fixed.

line 207: SAT hasn't been defined

Replaced with "surface temperature".

lines 270 and 275: "1998" -> "1997" (as year of a strong El Nino)

Fixed. Note that we've modified Fig. 4 so that the x-axis denotes the year of January in the DJF average. The text/caption are modified accordingly.

line 361: Quasi-biennial -> Quasi-Biennial

Fixed.

line 510: Figs. S1, S2 -> Figs. S4, S5

Thanks. These figures are now included in Appendix B (Figs. B8, B9), with text modified accordingly.

line 520 vs 185 vs 591: is it "best track", "Best Track" or "BestTrack"?

We now consistently used "Best Track".

line 528: should the 2 in kt2 be a superscript?

Yes, fixed.

line 556: suggest "multi-year skill" -> "multi-year skill or potential skill"

We have modified the sentence accordingly.

line 561: suggest to remove "obviously"

Done.

line 923: ACC map gross -> ACC map for gross

Fixed.

---

## Author Response (AR2)

Topical Editor comments are repeated below in italics followed by our response.

**Response to Review**

*Thank you very much for your revisions tag fully answer the referees' remarks. I was just wondering if there is a version number attached to SMYLE? If so, it should appear in the title. Also, I have a remark about the single vs plural form: e.g. at line 28, I think that the text between parentheses should use the plural form "~3-4 weeks", "~12 months", "~10 years". But at line 47, you should put "10-year duration" without the "s" as "10-year" qualifies "duration". Thank you for considering these additional minor remarks,*

> SMYLE does not have a version number (at this time). If and when a new version of SMYLE is created using the same model (CESM2), then clarification of the version will be appropriate. We've left the title as is. We've made edits in the text to address the single vs. plural form comments (call suggested changes accepted).